# Exploration and Enhancement of Classifiers in the Detection of Lung Cancer from Histopathological Images

**DOI:** 10.3390/diagnostics13203289

**Published:** 2023-10-23

**Authors:** Karthikeyan Shanmugam, Harikumar Rajaguru

**Affiliations:** Department of Electronics and Communication Engineering, Bannari Amman Institute of Technology, Sathyamangalam 638401, India; ms.karthi.1388@gmail.com

**Keywords:** histopathology, benign, adenocarcinoma, PSO, GWO, KL divergence, IWO, multilayer perceptron, bayesian linear discriminant analysis classifier

## Abstract

Lung cancer is a prevalent malignancy that impacts individuals of all genders and is often diagnosed late due to delayed symptoms. To catch it early, researchers are developing algorithms to study lung cancer images. The primary objective of this work is to propose a novel approach for the detection of lung cancer using histopathological images. In this work, the histopathological images underwent preprocessing, followed by segmentation using a modified approach of KFCM-based segmentation and the segmented image intensity values were dimensionally reduced using Particle Swarm Optimization (PSO) and Grey Wolf Optimization (GWO). Algorithms such as KL Divergence and Invasive Weed Optimization (IWO) are used for feature selection. Seven different classifiers such as SVM, KNN, Random Forest, Decision Tree, Softmax Discriminant, Multilayer Perceptron, and BLDC were used to analyze and classify the images as benign or malignant. Results were compared using standard metrics, and kappa analysis assessed classifier agreement. The Decision Tree Classifier with GWO feature extraction achieved good accuracy of 85.01% without feature selection and hyperparameter tuning approaches. Furthermore, we present a methodology to enhance the accuracy of the classifiers by employing hyperparameter tuning algorithms based on Adam and RAdam. By combining features from GWO and IWO, and using the RAdam algorithm, the Decision Tree classifier achieves the commendable accuracy of 91.57%.

## 1. Introduction

Cancer is increasingly common, and doctors use blood tests, biopsies, and image analysis for its diagnosis. It originates from damaged cells and varies among individuals. Understanding its source helps us comprehend its condition [1]. Lung cancer, often tied to smoking or harmful exposures, is a prevalent cancer type causing rising death tolls globally [2]. It affects both genders and has a low survival rate. Early detection is crucial for better outcomes. The five-year survival rate is approximately 34% for surgically removable early-stage cancer, compared to less than 10% for inoperable cases. Lung cancer treatment depends on histological characteristics, categorized as small cell (SCLC) and non-small cell (NSCLC) types, of which 80% to 85% are NSCLC and the rest are SCLC [3]. NSCLC has subtypes such as benign, adenocarcinoma (ACA), and squamous cell carcinoma (SCC). SCC displays characteristics such as the presence of clusters of polyhedral cells, keratinization, and the formulation of keratin pearls. Once the tissue type is identified, suitable treatments can be selected: either surgery, chemotherapy, radiation, targeted therapy, or immunotherapy.

Early detection and treatment of cancer are vital for better patient outcomes. Traditional diagnostic methods involve clinical assessments, lab tests, imaging, and a procedure called biopsy [4], which is considered the gold standard. During biopsy, tissue samples are taken and examined under a microscope using techniques such as hematoxylin and eosin staining. This histopathological analysis helps identify abnormal tissue growth and cell characteristics. Accurate identification and classification of individual cell nuclei are of utmost importance when evaluating tissue samples for cancer diagnosis. Pathologists inspect these samples at different magnifications, looking for signs of malignancy such as irregular cell shape, dark nuclei, and increased mitotic figures [5], and count to generate reliable results [6]. Manual histopathological examination is a time-consuming process due to the frequent presence of numerous nuclei from diverse categories clustered together in histopathological images, which can result in disagreements among pathologists [7], prompting the development of automated systems. Researchers have utilized image processing, pattern recognition, and machine learning/deep learning techniques to create computer-aided diagnostic (CAD) systems. These systems aim to detect and classify carcinomas quickly and reliably [8]. Machine learning and deep learning algorithms improve CAD performance as they learn from more data. These approaches use either microscopic images or whole slide images (WSIs) and extract features to aid in diagnosis. The challenge is to create a novel, versatile, and fully automated CAD system that can handle both microscopic images and WSIs, regardless of any imaging artifacts. Automated analysis of microscopic images is vital for evaluating digitized specimens, reducing inter-observer variations, and improving objectivity and reproducibility, as emphasized by Foran et al. [9]. This advancement can enable comparative studies of diseases and potentially aid in diagnostic decision-making.

Different imaging techniques, such as ultrasounds, MRIs, CT scans, X-rays, and needle biopsies, are used to diagnose lung cancer. X-ray imaging, considered a fundamental technique for lung examination, possesses restricted resolution and the potential to overlook specific areas of interest [10]. CT scans are commonly used to detect early stages of lung cancer and locate tumors before surgery, but they expose patients to harmful radiation with repeated scans. MRI demonstrates notable sensitivity and specificity, valuable for identifying bone metastases, although it is not advisable for diagnosing lung cancer. Ultrasound, a non-invasive method, proves adept at identifying postoperative lung issues and surpasses X-rays in effectiveness [11]. While image examination aids in diagnosis, staging, treatment evaluation, and prognosis assessment, histopathological examination remains the most reliable method to determine tumor characteristics and clinical stages. Histopathological images offer an intricate view of cellular and tissue-level transformations linked in differentiating between various conditions and cancer types, empowering pathologists to deliver precise and reliable diagnoses. Moreover, they are invaluable for pinpointing distinct biomarkers linked to various cancer types and grades, facilitating tumor classification and subtyping. Histopathological images form a dependable diagnostic framework known for its consistency and reliability in cancer diagnosis [12]. By harnessing extensive datasets of annotated histopathological images, it becomes feasible to create highly dependable algorithms for automated cancer diagnosis. These algorithms effectively streamline the diagnostic process, reducing the necessity for extensive manual examination [13].

The objective of this study is to create a classification framework that can analyze histopathological images data related to lung cancer. The goal is to accurately classify individuals as either having cancer or not, using machine learning techniques and meta-heuristic algorithms for tasks such as feature extraction, feature selection, and classification. The following subsection analyzes various methods for cancer detection and classification using image processing and classification techniques.

### Review of Previous Work

In recent times, the research community has shown significant interest in diagnosing Lung Cancer through histopathological images. Numerous methodologies have been explored, utilizing a range of machine learning and deep learning techniques, across diverse datasets to detect instances of lung cancer.

Various strategies have been proposed to identify irregularities in lung-related images, encompassing chest radiographs, CT scans, ultrasound images, histopathological images, and microarray data. Ozekes and Camurcu [14] utilized template matching, while Schilham et al. devised a computer-aided detection (CAD) system that encompasses preprocessing, the identification of candidate nodules, feature extraction, and cancer classification [15]. Wang et al. [16] executed the classification of pathology images concerning lung cancer using a convolutional neural network (CNN) methodology, incorporating cell segmentation. The final layer of the CNN model integrated the Softmax activation function to enhance the classification process. Through the application of the region of interest (ROI) technique as a preliminary step, they focused on cell areas containing relevant tumors. The achieved classification accuracy for the three-class image dataset reached 90.1%. Dehmeshki et al. [17] employed a genetic algorithm based on shapes for template matching, while Suarez-Cuenca et al. used an iris filter for CT image discrimination [18]. Murphy et al. used a K-nearest neighbor (KNN) classifier for nodule detection [19], and Giger et al. used geometric features in their CAD system for CT images.

Wei et al. [20] undertook the categorization of histopathological images depicting six classes of lung cancer utilizing CNNs. They specifically employed ResNet models for their investigation. The ResNet models were integrated with pre-trained approaches from ImageNet and COCO image databases. Prior to the model training phase, the input data underwent preprocessing, which included the application of augmentation techniques. The study’s achievement in terms of classification F-score reached a notable 90.4%. Mohammed Al-Jabber et al. [21] employed histopathological images from the LC25000 dataset, employing both ANN and the GoogLeNet and VGG-19 models. This combination yielded an impressive accuracy of 99.64%. Teramoto et al. [22] effectively distinguished histopathological images spanning three types of lung cancer through the application of a deep learning model. They implemented an augmentation approach that involved rotating, flipping, and applying filters to each image. Following this, they employed their developed deep CNN model to carry out the classification process. The outcomes of their classification efforts yielded an accuracy of around 70%. Shapcott et al. [23] conducted their model training by initially subjecting the input data to a preprocessing stage, integrating the augmentation technique. They employed a deep learning methodology for classifying histopathological images related to colon cancer. The dataset encompassed four distinct classes. To facilitate cell identification, a cell patches algorithm was employed on each image. The images were segmented into specific dimensions through segmentation procedures. The classification process was then conducted using the CNN model based on the defined cell patches. The obtained correlation accuracy rates ranged between 90% and 96.9%.

Barker et al. an automated system to classify brain tumors using digital pathology images [24]. Ojansivu et al. explored an automated method for categorizing breast cancer from tissue samples [25]. Ficsor et al. proposed an automated classification method for colon inflammation using digital microscopy images of histological sections [26]. The authors of a study, Mouelhi et al. [27], used various techniques such as Haralick’s textures, histogram of oriented gradients (HOG), and color-based statistical moments (CCSM) to extract features from biopsy images and classify cancerous cells. The features included energy, correlation, homogeneity, contrast, GLCM texture features, as well as RGB, gray level, and HSV color components. Huang and Lai [28] focused on histology image analysis, employing texture features and KNN, SVM for image classification and segmentation. Their approach achieved a classification accuracy of 90.07% and 92.8%. Gessert et al. [29] executed the classification procedure employing CNN models based on transfer learning, leveraging microscopic images of colon cancer. Their study employed a dataset that comprised both benign and malignant images. They trained the dataset using various models including Inception, VGG, and DenseNet. Among these, the DenseNet model yielded the most promising classification outcome, achieving a classification accuracy of 91.2%.

Sinha and Ramkrishan [30] studied small biopsy images, analyzing cell characteristics such as shape, size, color, and other properties. Four classification methods were compared: Bayesian, KNN, neural networks, and SVM. The last two methods achieved the highest accuracy rates of 94.1%, while the first two had lower rates of 82.3% and 70.6%. Kasmin et al. [31] examined microscopic biopsy images, considering characteristics such as cell/nuclei size, cell boundary length, minimum polygon area enclosing a cell, major axis length of an ellipse fitted to a cell, filled cell area, and average cytoplasmic intensity. They used neural networks and achieved classification accuracies of 86% and 92%. Chia-Hung Chen et al. [32] used a convolutional neural network to diagnose endobronchial ultrasound images, achieving an improved accuracy of 85.4% compared to traditional methods. Azka Khoirunnia et al. [33] developed a lung cancer detection system using a combination of CNN and RNN with Microarray data. In their research, CNN achieved 83% accuracy, RNN reached 71%, and the fusion of CNN and RNN (CRNN) attained the highest accuracy at 91%. Shahid Mehmood et al. [34] focused on classifying histopathological images of lung and colon cancers. By using AlexNet along with a technique called Class-Selective Contrast Enhancement, they achieved an impressive accuracy of 98.4%.

This paper is structured as follows: Section 2 focuses on the methodology employed for detecting lung cancer. Section 3 explores the feature extraction techniques including Particle Swarm Optimization and Grey Wolf Optimization whereas Section 4 explores the feature selection techniques, such as KL Divergence, and Invasive Weed Optimization. Section 5 explains the different classifiers used and hyper parameter updating method and its implementation. Section 6 presents the cumulative results, and Section 7 concludes the paper.

The following section deals with the methodology employed for identifying lung cancer through histopathological images.

## 2. Methodology for Lung Cancer Detection

This study employed lung histopathological images sourced from the LC25000 Dataset, which is available online. Andrew Borkowski and his colleagues from James Hospital Tampa, University of South Florida, and the Moffitt Cancer Center in Florida, USA, worked together to collectively assemble this dataset. The dataset encompasses histopathological images representing lung and colon cancer cases. Excluding colon cancer cases, the collection includes a total of 500 lung tissue images, divided equally between Benign Lung tissue and Lung Adenocarcinomas. These images were originally captured from pathology glass slides and were later resized to square dimensions of 768 × 768 pixels, down from their original size of 1024 × 768 pixels. The dataset underwent augmentation, resulting in an expansion to a comprehensive set of 10,000 lung histopathological color images which are categorized into two classes: Benign (N) and Adenocarcinoma (ACA), each consisting of 5000 images. These images are resized to a standard size of 256 × 256 followed by converting into a grey scale image. Notably, the images portray lung benign tissue characterized by abnormality but not indicative of cancer, while lung adenocarcinoma, the most prevalent form of lung cancer in the United States and notably linked to smoking, forms the second category.

Figure 1 shows the general schematic diagram for identifying and categorizing lung cancer in histopathological images. The input histopathological image will undergo conversion into a linear vector comprising 65,536 elements (due to the image’s size of [256 × 256]). The procedure involves image pre-processing and a modified KFCM-based segmentation. During segmentation process approximately as [190 × 190] of the original image (i.e., nearly 36,100 intensity values) are segmented and used for further processing. These values will be directly employed to initialize the positions of birds in the Particle Swarm Optimization (PSO) and grey wolves in the Grey Wolf Optimization (GWO) algorithms. Optimization algorithms such as PSO and GWO are used to obtain a matrix of [512 × 10] dimensionally reduced intensity values from the segmented images. These dimensionally reduced features undergo feature selection techniques such as KL divergence and IWO. The selected features are then inputted into classifiers to evaluate their performance of the classifiers. Furthermore, an enhancement in the accuracy of lung cancer classification across various classifiers including SVM, KNN, Random Forest, Decision Tree, Softmax Discriminant, Multilayer Perceptron, and BLDC classifiers is achieved through the implementation of a Hyper Parameter Updation algorithm based on the RAdam technique.

### 2.1. Histopathological Image Preprocessing

Histopathological analysis serves as the definitive standard for evaluating the quality and clinical staging of tumors [35]. In the realm of diagnosing and treating medical conditions, healthcare professionals heavily rely on histopathological images. These images establish a crucial cornerstone for predicting patient survival rates [36].

As per available reports, histopathological images present several challenges:The images exhibit intricate geometric structures and complex textures that arise from the vast diversity in structural morphology [37].Notably, histopathological images are susceptible to color inconsistencies and noise due to external factors such as variations in illumination conditions [38].Variations in microscope magnification, equipment settings, and other variables contribute to inconsistencies in image sizes and resolutions within histopathological images [39].Elements of significance, such as local micro-vessels with distinctive textural characteristics, significantly influence disease diagnosis within histopathological images. Extracting these features is of paramount importance in supporting the classification and diagnosis of lung cancer [40].

Due to these factors, the histopathological images we encounter are frequently not perfect and these images show that image quality is affected by noise during acquisition and artifacts during sample preparation and slide digitization. Preprocessing methods are employed in histopathological images to enhance image quality, rectify anomalies, amplify pertinent characteristics, and establish uniformity, ultimately resulting in heightened precision and dependability of diagnostic outcomes. The study demonstrates that using an efficient adaptive median filter enhances image quality, reduces artifacts, and facilitates accurate diagnosis and analysis. However, when subjected to an adaptive median filter, these images tend to become smoother and exhibit reduced noise, rendering them suitable for our forthcoming investigations. After artifact removal, the filtered histopathological images are used for segmentation. Here, the size of the selected region of interest (ROI) is 256 × 256 which is the complete original image.

### 2.2. Histopathological Image Segmentation

A Modified Kernel Fuzzy C-Means methodology is employed to effectively segment normal and abnormal regions in histopathological images even though outliers are encountered. Image segmentation is the process of dividing an image into distinct regions based on certain image characteristics, with the goal of isolating and identifying specific regions within the image [41]. In this scenario, we have an input histopathological image denoted as H, which consists of a set of color images $x_{i}$ at pixel i (i=1, 2, … N) and these color images are represented as $X=\left\{x_{1},\;x_{2},\;\ldots ,\;x_{N}\;\right\}\;⸦\;R^{k}$, in the k-dimensional space. The cluster centers within the histopathological images are represented as $Y=\left\{y_{1},\;y_{2},\;\ldots ,\;y_{c}\;\right\}$, where c is a positive integer $\left(2<c\ll N\right)$, and $m_{ij}$ represents the membership value for each pixel i in the j-th cluster (j=1, 2, … c). In the Kernel Fuzzy C-Means algorithm, clusters are formed in the image space by assigning distinct membership values to all pixels. The objective function or general equation for the Kernel Fuzzy C-Means algorithm is expressed as follows in the Equation (1):(1)$$O_{KFCM}=\sum_{i=1}^{N}\sum_{j=1}^{c}m_{ij}^{n}\;{\left\|x_{i}-y_{j}\right\|}^{2},\;1\;\leq n\leq \infty$$
where *n* represent an exponent used for regularization, with the condition that, *n* > 1, and ${\left\|x_{i}-y_{j}\right\|}^{2}$ denotes the squared grayscale Euclidean distance between $x_{i}$ and $y_{j}$, which is given in Equation (2):(2)$$\sum_{j=1}^{c}m_{ij}=1\;,\;\;m_{ij}\;\in \;\left[0,1\right]\;,\;\;0\leq \sum_{i=1}^{N}m_{ij}\leq N$$

Using the membership function derived from the alternate optimization approach, the process of iteratively updating the cluster centers is carried out according to the Equations (3) and (4).
(3)$$m_{ij}=\frac{1}{\sum_{k=1}^{c}{\left({\left\|x_{i}-y_{j}\right\|}^{2}/{\left\|x_{i}-y_{k}\right\|}^{2}\right)}^{1/\left(n-1\right)}}$$(4)$$y_{j}=\frac{\sum_{i=1}^{N}m_{ij}^{n}x_{i}}{\sum_{i=1}^{N}m_{ij}^{n}}$$

To reduce the impact of noise, the Equation (5) incorporates the spatial information of neighboring pixels,
(5)$$O_{KFCM-S}=\sum_{i=1}^{N}\sum_{j=1}^{c}m_{ij}^{n}\;{\left\|x_{i}-y_{j}\right\|}^{2}\;+\frac{\alpha }{N_{R}}\sum_{i=1}^{N}\sum_{j=1}^{c}m_{ij}^{n}\;\left(\sum_{r\in N_{i}}{\left\|x_{i}-y_{j}\right\|}^{2}\right)$$

Here, the spatial information is denoted by α, $N_{i}$ represents the set of pixels and its cardinality is defined as $N_{R}$, the neighborhood function is substituted by ${\left\|\overset{´}{x_{i}}-y_{j}\right\|}^{2}$, in place of $\frac{1}{N_{R}}\sum_{r\in N_{R}}{\left\|x_{i}-y_{j}\right\|}^{2}$, where, $\overset{´}{x}$ represents a color scale-filtered image, and the Euclidean distance is replaced with the correlation distance measure to avoid the neighborhood function. The updated equation is represented in Equation (6):(6)$$O_{KFCM-S(1,2)}=\sum_{i=1}^{N}\sum_{j=1}^{c}m_{ij}^{n}\;{\left\|x_{i}-y_{j}\right\|}^{2}\;+\alpha \sum_{i=1}^{N}\sum_{j=1}^{c}m_{ij}^{n}\;\left({\left\|x_{i}^{\prime }-y_{j}\right\|}^{2}\right)$$

In this study, a modified version of KFCM computes the parameter $\eta_{j}$ for each cluster at every iteration to substitute for α [42]. The calculation of this parameter utilizes the correlation function, as outlined in the Equation (7):(7)$$\eta_{j}\;=\;\frac{\min_{j^{\prime }\neq j\;}\left(1-C\left({y^{\prime }}_{j},y_{j}\right)\right)}{\max_{k\;}\left(1-C\left(y_{k},x^{\prime }\right)\right)}$$

Here *C* represents the correlation function or correlation distance measure. Here, determining the precise characteristics of *C* typically necessitates a large number of patterns and numerous cluster centers to identify optimal value for $\eta_{j}$. To address this challenge, a solution is devised by integrating spatial context and scale information through the incorporation of fuzzy factor. The objective function of the KFCM, as presented in Equation (8), incorporates the inclusion of the fuzzy factor $F_{ij}$.
(8)$$O_{C-KFCM}\;=\;\sum_{i=1}^{N}\sum_{j=1}^{c}\left[m\;\left|ij^{n}\right|\;{\left\|x_{i}-y_{j}\right\|}^{2}+F_{ij}\right]$$

Then the altered fuzzy factor $F_{ij}^{\prime }$ is derived using Equation (9).
(9)$$F_{ij}^{’}\;=\;\sum_{C\in N,\;i\neq k}w_{ik}{\left(1-m_{ij}\right)}^{m}$$

This adjusted fuzzy factor plays a crucial role in influencing local neighbor relationships and substituting the traditional distance metric with a correlation function. Here, $w_{ik}$ represents the fuzzy factor for cluster i, and $1-C\left(x_{i}-y_{j}\right)$ signifies the correlation metric function. Since the histopathological images contain variation in intensities, gradients, and complex backgrounds, it becomes imperative to employ a modified KFCM-based segmentation method to distinguish between the region of interest (ROI) and the background in the image. Figure 2 illustrates the sequence of the original image, the filtered image, the identified ROI within the ACA image, and the segmented image generated using the modified KFCM for the Adenocarcinoma (ACA) class.

The following section focuses on the methods utilized for extracting dimensionally reduced image features, aimed at enhancing the classification and detection of lung cancer using histopathological images.

## 3. Feature Extraction

Feature extraction techniques condense essential information from images into compact feature vectors, enabling the effective classification of complex image datasets using linear algorithms [43]. As the abundant features within histopathological images serve as a fundamental resource for clinicians to conduct diagnoses, the proficient extraction of these image features stands as a pivotal factor in enhancing the precision of computer-aided diagnosis [44]. This study delves into the impact of two distinct feature extraction techniques such as PSO and GWO on the classification of histopathological images related to lung cancer.

### 3.1. Particle Swarm Optimization (PSO)

Kennedy and Eberhart introduced the PSO algorithm in 1995, which draws inspiration from the hunting behavior of birds. This optimization method relies on a population and leverages the social dynamics of bird flocks. It starts by creating particles and setting key parameters for the optimization process. [45].

Every particle has a unique position that is traced by the following equation:(10)$$x_{i}^{k}=\left(x_{i1}^{k},x_{i2}^{k},\ldots ,x_{iq}^{k}\right)$$

The velocity is traced by the following equation:(11)$$y_{i}^{k}=\left(y_{i1}^{k},y_{i2}^{k},\ldots ,y_{iq}^{k}\right)$$

Each particle’s velocity is updated as:(12)$$y_{i}^{k+1}=w_{i}y_{i}^{k}+c_{1}r_{1}\left(pbest_{i}-x_{i}^{k}\right)+c_{2}r_{2}\left(gbest_{i}-x_{i}^{k}\right)$$

Here, $r_{1}$ and $r_{2}$ represent randomly selected values within the range of 0 to 1. The acceleration coefficients, denoted as $c_{1}$ and $c_{2}$, play a role in analyzing the motion of particles. The weight function is expressed as:(13)$$w_{i}=\frac{w_{max}-w_{min}}{k_{max}}\times k$$

The position of each particle is given by:(14)$$x_{i}^{k+1}=x_{i}^{k}+y_{i}^{k+1}$$

The particle that possesses the optimal position progresses to the next level. The best position for an individual particle is represented by the letters “p-best”, while the letters “g-best” represent the best position among all particles. The weight parameter “$w_{i}$” is chosen between 0.45–0.9, maximum iteration values are 100–1000, both $r_{1}$ and $r_{2}$ are set to 0.85, cognitive component ($c_{1})$ and Social Component ($c_{2})$ are chosen between 1.0–2.0. The above values are determined based on the trial-and-error method.

### 3.2. Grey Wolf Optimization (GWO)

Grey wolves are known for living and hunting in groups called packs [46]. The process of searching and hunting involves plotting to track and approach a target efficiently. This optimization technique, inspired by the search and hunting patterns of gray wolves, employs symbols such as Alpha (α), Beta (β), and Gamma (γ) to represent the best, next best, and third best solutions in mathematical modeling. Lambdas are presumed to be the remaining possible solutions and they guide the alpha, beta, and gamma wolves in searching and surrounding the prey. Three coefficients, A, B, and C are suggested to describe the encircling behavior. The equation of hunting strategy is formulated as follows:(15)$$D_{\alpha }=\left|B_{1}\cdot X_{\alpha }-X(t)\right|$$
(16)$$D_{\beta }=\left|B_{2}\cdot X_{\beta }-X(t)\right|$$
(17)$$D_{\gamma }=\left|B_{3}\cdot X_{\gamma }-X(t)\right|$$
where $D_{\alpha },D_{\beta }$ and $D_{\gamma }$ denotes the adjusted distance variables from the alpha, beta, and delta positions to the other wolves, $B_{1}$, $B_{2}$ and $B_{3}$ are coefficients that assist in adapting these distance variables, *t* signifies the ongoing iteration, X indicates the position of the grey wolf and it follows as,
(18)$$X_{1}=\left|X_{\alpha }-A_{1}D_{\alpha }\right|$$
(19)$$X_{2}=\left|X_{\beta }-A_{2}D_{\beta }\right|$$
(20)$$X_{3}=\left|X_{\gamma }-A_{3}D_{\gamma }\right|$$
(21)$$X(t)=\frac{X_{1}+X_{2}+X_{3}}{3}$$

The parameters A and B can be mathematically expressed as follows:(22)$$A=2i\cdot r_{1}-i$$
(23)$$B=2\cdot r_{2}$$

The control parameter  i  chases A, which eventually drives the lambda wolves to flee from the dominant wolves such as α, β and γ. When there are multiple dominant wolves (|A| > 1), the grey wolves run away from them, allowing lambda wolves to search extensively and explore more during optimization. However, when there are fewer dominant wolves (|A| < 1), the grey wolves approach them and follow their guidance in hunting, which is called local search in optimization. During the iterations, the control parameter *i* is linearly decreased from 2 to 0, and is represented as,
(24)$$i=2-(iter)\cdot \frac{2}{\max\_iter}$$
where max_iter indicates the maximum iteration, and it is started from the beginning.

In the context of the classification problem, the introduction of randomness through variables $r_{1}$ and $r_{2}$ leads to heightened fluctuations in the wolves’ positions. Consequently, their ability to effectively converge towards the target (prey) becomes hindered. To address this issue, a decision has been made to treat the values of $r_{1}$ and $r_{2}$ in Equations (9) and (10) as control parameters within a confined range of [0, 1], rather than allowing them to remain purely random. Through empirical experimentation, it has been determined that the optimal performance of the Grey Wolf Optimization (GWO) algorithm is achieved when both $r_{1}$ and $r_{2}$ are set to 0.8. This adjustment enhances the accuracy of the GWO algorithm in tackling the classification problem.

### 3.3. Statistical Analysis

To enhance the accuracy of cancer prediction using dimensionally reduced features, it is advisable to calculate statistical parameters from the region of interest. The intensity values, which have been reduced in dimensionality through methods such as PSO (Particle Swarm Optimization) and GWO (Grey Wolf Optimization), are then examined using statistical measures such as Mean, Variance, Skewness, Kurtosis, Pearson Correlation Coefficient (PCC), and CCA (Canonical Correlation Analysis). These statistical parameters help determine whether the outcomes accurately reflect the inherent properties of lung cancer data within the subspace. These attributes were derived for both normal and malignant classes.

The statistical parameters of cancer data, extracted using the PSO and GWO methods, are shown in Table 1. Variance quantifies data spread. Notably, Table 1 reveals lower mean values for normal cases using both PSO and GWO, while higher mean values are evident for malignant cases using both methods. Furthermore, the Malignant group demonstrates greater data spread compared to the Normal group as indicated by Table 1. GWO shows a Pearson correlation coefficient of 1 for both cases, implying strong intra-class correlation. Skewness and kurtosis are highly skewed for both normal and malignant instances. When CCA values exceed 0.5, strong inter-class correlation is present. However, Table 1 indicates that PSO and GWO methods exhibit the lowest inter-class correlation. Consequently, the analysis of these extracted features emphasizes the need for improved classifiers.

In cases where the features exhibit linear separability, a straightforward binary thresholding approach can be employed for the classification of Histopathological Lung images into two distinct classes: N and ACA. The characteristics of malignancy exhibit non-linear and non-Gaussian features that overlap with each other. To analyze these dimensionally reduced values which was obtained from PSO and GWO methods, histogram and scatterplot plots are used as illustrated in Figure 3 and Figure 4.

The histogram plot in Figure 3 illustrates the distribution of PSO feature data for normal and malignant cancer cases. The histogram illustrates PSO features characterized by outliers, substantial gaps, downward trends, and a non-Gaussian distribution. From Table 1, In the PSO-based extraction technique, the Canonical Correlation Coefficient (CCA) value is significantly low at 0.12309, suggesting a non-linear relationship between normal and malignant cases. Figure 4 showcases the histogram plot for GWO feature distribution, indicating skewed Poisson distributed data, and a non-linear nature.

Figure 5 and Figure 6 display scatterplots demonstrating the feature output of normal and malignant cancer data utilizing the PSO and GWO methods. Scatter plots are useful for identifying data clustering, detecting nonlinearity, and overlapping. Both figures indicate the presence of nonlinearity and overlapping in the data. Therefore, from the histogram and scatterplot it is evident to employ accurate classifiers capable of distinguishing between normal and cancer cases in lung data using PSO and GWO features. The next section centers on the techniques applied to choose optimal image features, with the goal of improving the classification and identification of lung cancer in histopathological images.

## 4. Feature Selection

Feature selection aims to reduce input variables, excluding irrelevant characteristics for a more accurate, less complex, and unbiased model. Optimal feature selection is crucial for creating an effective, accurate machine learning model with high generalization ability [40]. In this paper, Feature Selection is performed using the KL Divergence and Invasive Weed Optimization (IWO) methods. Following the feature extraction procedures as described in Section 3, which involves Particle Swarm Optimization (PSO) and Grey Wolf Optimization (GWO). After feature extraction method [256 × 256] is dimensionally reduced to [512 × 10] per histopathological image. These [512 × 10] intensity values per image serve as the initial input for the Feature Selection techniques namely KL Divergence and IWO. However, the application of Feature Selection techniques is a further dimensionality reduction method only. After PSO and GWO feature extraction methods through KL Divergence and IWO process [512 × 10] intensity values per image is reduced to [100 × 12] which represents the most relevant intensity values of the images are retained as input given to the classifier for the subsequent classification process. The histopathological images are represented as a relevant intensity value in matrix form as described above.

### 4.1. KL Divergence

KL Divergence, also known as relative entropy, measures disparities between probability distributions, but in an asymmetric manner. The KL divergence between a probability distribution $q=(q_{1},q_{2},\ldots ,q_{n})$ and another distribution $p=(p_{1},p_{2},\ldots ,p_{n})$ is defined as,
(25)$$D_{KL}(q||p)=\sum_{j=1}^{m}q_{j}\log\frac{q_{j}}{p_{j}}$$

The integral form of the *KL* divergence for continuous distributions is expressed as follows:(26)$$D_{KL}(q||p)=\int_{-\infty }^{\infty }q_{j}\log\frac{q_{j}}{p_{j}}dx$$

The *KL* divergence exhibits mutual convexity for both discrete and continuous distributions. The following are the properties of the *KL* divergence measure:(27)$$D_{KL}(q||p)=\left\{\begin{array}{c} 0,\;if\;q=p\\ c,\;c>0,\;if\;partially\;overlapping\\ +\infty ,\;if\;non-overlapping \end{array}\right.$$

From the above equation, it can be observed that when the *KL* divergence is smaller, the two compared distributions are more similar.

### 4.2. Invasive Weed Optimization

The invasive weed optimization algorithm is a popular population-based metaheuristic approach [47]. The dynamic and versatile characteristics of weed colonies have sparked the creation of an optimization algorithm that imitates their behavior. By leveraging the qualities of weeds, a straightforward and efficient optimization technique can be developed. This method, called the IWO algorithm, incorporates phases such as seeding, growth, and competition. The following are the strategy for simulating weed habitat behavior:Primary Population Initialization: A few seeds are dispersed to start the search.Reproduction process: Seeds have the potential to grow into flowering plants, which then choose and spread the fittest seeds for survival and reproduction. The quantity of grass grain grains decreases in a linear fashion from $Y_{max}$ to $Y_{min}$ as follows:(28)$$n(weed_{j})=\frac{Y_{\max}(\max\_fit-fit(weed_{j}))+Y_{\min}(fit(weed_{j})-\min\_fit)}{\max\_fit-\min\_fit}$$Spectral Spread Method: The group’s seeds are distributed normally with a mean planting position and standard deviation (SD) determined by the equation below.
(29)$$\sigma_{t}={\left(\frac{N-t}{N}\right)}^{m}\left(\sigma_{\mathrm{int}ital}-\sigma_{final}\right)+\sigma_{final}$$Competitive Deprivation: If the colony has more grasses than the maximum limit (Smax), the grass with the lowest fitness is eliminated to maintain a consistent number of herbs.The process continues until the maximum iteration is reached, keeping the lowest cost value of the grasses.

The upcoming next section revolves around the utilization of classification methods to categorize lung cancer images within histopathological images.

## 5. Classifiers for the Detection of Lung Cancer

Classifiers have a crucial role in categorizing data effectively. An optimal classifier is characterized by its ability to achieve high accuracy and low error rates while maintaining manageable computational complexity. Addressing the classification challenge involves constructing a model for the purpose of classifying images and assigning them appropriate class labels. The following sections of this paper delve into the classifiers that were used for this purpose.

### 5.1. Support Vector Machine

SVM is known for its scalability and classification performance [42]. It aims to create a hyperplane that maximizes class separation by minimizing the cost function. It is given by the following expression:(30)$$Minimize,\;\frac{1}{2}\;{\left\|w\right\|}^{2}+C\;\sum_{k=1}^{m}\mu_{k}$$
$$\mathrm{Subject}\;\mathrm{to}\;z_{k}{\left(w^{T}x_{k}+f\right)}^{3}\geq 1-\mu_{k},\mu_{k}\geq 0$$
where $w^{T},x_{k}\in R^{2}$ and ${f\in R,{\left\|w\right\|}^{2}=w}^{T}w$.

*C* represents the trade-off between the margin and the error. The training data’s size is represented by $\sum_{k}$, and the class label for each sample is represented as $z_{k}$. SVM is a flexible classifier suitable for linear and nonlinear cases. To handle nonlinear data, we employ Polynomial, RBF, and Sigmoid kernel functions. In this study, we exclusively enhance the classification accuracy by utilizing the SVM-RBF kernel.

### 5.2. K-Nearest Neighbor

KNN stands as a widely utilized and efficient non-parametric classification technique. In KNN, the symbol ‘k’ denotes the count of nearest neighbors involved in the voting process. To enhance prediction accuracy, employing an odd value for k is recommended. KNN determines the classification of a test sample by conducting a majority vote among neighboring training samples. Measuring distances between individuals is crucial, and the Euclidean distance is commonly used for this purpose [48]. For example, in the Euclidean space if u and v are the two points and it is assumed that $u=(u_{1},u_{2},u_{3},\ldots ,u_{n})$ and $v=(v_{1},v_{2},v_{3},\ldots ,v_{n})$, then the Euclidean distance of line segment can be expressed as follows:(31)$$Dist(u,y)=\sqrt{(u_{1}-v_{1}{)}^{2}+(u_{2}-v_{2}{)}^{2}+\ldots +(u_{n}-v_{n}{)}^{2}}=\sqrt{\sum_{i=1}^{n}(u_{i}-v_{i}{)}^{2}}$$

### 5.3. Random Forest

This tree-based ensemble learning algorithm is highly accurate and resilient in image classification [49]. It utilizes multiple decision trees that work independently. Two important parameters for the algorithm are the number of decision trees and the number of predictive variables used in each tree’s decision-making process. By combining the votes of multiple decision trees, a random forest can accurately predict binary tasks. For a training set × consisting of M samples, each containing N features and a classification label Y. The following steps are involved in the construction of Random Forest.

Randomly select M samples from × using the Bootstrap method.Choose *n* random features (where *n* < N) to split a decision tree node. Determine the split criterion by selecting the feature with the lowest Gini value. Gini is computed using the formula:(32)$$Gini=1-\sum_{i=1}^{c}(p_{i}{)}^{2}$$
where $p_{i}$ represents the relative frequency of dataset features and c represents the number of classes.Generate M decision trees by repeating steps 1 and 2, M times.Create a random forest by combining the decision trees and utilize voting to determine the classification outcome.

### 5.4. Decision Tree

It is a well-known machine learning algorithm that partitions input data recursively [50]. A decision tree starts with a root node and branches. This work utilizes CART, which splits the data based on its ability to distinguish between groups. The process continues until all data groups have the same label or match the training set. CART uses the Gini impurity measure at each node to determine the best split. The data at node ‘d’ are divided into two subsets, X-left and X-right, based on the splitting features and a threshold determined by CART and the amount of data X.

At node ‘d’ the input is computed through impurity measure Gini as $\;\sum_{k}p_{dk}\left(1-p_{dk}\right)$ with the proportion of class k observation in the node ‘d’. Construction time of a decision tree depends on the dataset’s size (samples and features). Overfitting can occur if the tree is built using CART and results in few samples per leaf. To prevent overfitting and improve accuracy, a pruning algorithm can be used to simplify the tree, reducing construction time while maintaining performance.

### 5.5. Softmax Discriminant Classifier

SDC’s main objective is to classify a given test specimen [51] by comparing its distance to the training sample within its category. The process entails gauging the distance between training and test samples belonging to the same class to derive the outcome. Supposing, the training set $M=\left[M1,\;M2,\ldots ,\;Mq\right]\in R^{c\times d}$ comes from q distinct classes. $M_{q}=\left[{M1}^{q},\;{M2}^{q},\ldots \;{Mdq}^{q}\right]\in R^{c\times d_{q}}$ Indicates dq samples from the qth class where $\sum_{j=1}^{q}d_{j}=d$. Assuming $w\epsilon R^{c\times 1}$ represents the test sample, within the classifier, we employ samples from class q to recognize the test sample, aiming to minimize the reconstruction error. To uphold the principle of SDC, we can enhance the non-linear transformation linking the q class samples and the test sample. Therefore, the SDC can be defined as follows:(33)$$h\left(w\right)=\arg\underset{j}{\max}z_{w}^{j}$$
(34)$$h\left(w\right)=\arg\underset{j}{\max}\log\left(\sum_{j=1}^{d_{j}}\exp\left(-\lambda \;{\left\|{w-w}_{k}^{j}\right\|}_{2}\right)\right)$$
where $h\;\left(w\right)$ defines the distance between the jth class and the test sample. The value of 𝜆 should be greater than zero, to provide a penalty cost. If w relates to the jth class, then w and ${\;w}_{k}^{j}$ would have likely same characteristics and so ${\left\|{w-w}_{k}^{j}\right\|}_{2}$ is progressing close towards zero and hence maximizing $z_{w}^{j}$ can achieve the maximum possible value in an asymptotic manner.

### 5.6. Multilayer Perceptron

MLP is often used to approximate functions such as regression [52]. It consists of an input layer with *n* nodes, a hidden layer, and an output layer. The given input and output pairs be denoted as ${(m}_{p},\;n_{p}),\;p=1,\;2,\;\ldots ,\;m,\;$ where $m_{p}=(m_{p1},\;m_{p2},\;\ldots ,\;m_{pn})$ and $y_{p\;}$ are the input vector and the corresponding desired output value, respectively. Sigmoid function is commonly used for hidden and output nodes, producing values from 0 to 1.

The kth hidden node in the MLP calculates its output when the input is given. The output value is computed as
(35)$$c_{pk=}{\;f}_{s}\;\left(\sum_{j=1}^{n}w_{jk}m_{pj}+\theta_{k}\right)$$

The output value of the output node is determined by the sigmoid function ${(f}_{s})$, along with bias $(\theta_{k}$), and connection weight ${(w}_{jk})$ associated with the corresponding hidden node. Then the final output value is computed as,
(36)$$c_{p=}\;f_{s}\;\left(\sum_{k=1}^{l}w_{k}c_{pk}+\theta \right)$$

The number of hidden nodes is denoted by l, the bias to the output node is represented by 𝜃, and $w_{k}$ signifies the connection weight from the $k^{th}$ hidden node to the output node. This results in a total of $\left(n+2\right)l+1\;$ synaptic connections. To train the Multilayer Perceptron (MLP), the following cost function can be utilized.
(37)$$E=\frac{1}{2}\;\sum_{j=1}^{t}{\left(n_{p}-c_{p}\right)}^{2}$$
where *t* denotes the number of training patterns. In our study, we used a three-layer model, which is known to effectively approximate any continuous function with high accuracy [53].

### 5.7. Bayesian Linear Discriminant Classifier

The BLDC, or Bayesian Linear Discriminant Classifier, can distinguish between multiple classes. It uses the Fisher linear discriminant and applies the Bayes decision rule to estimate the error probability [54]. Bayesian regression assumes that the target variable y is a linear combination of vector k, and Gaussian noise m. This relationship is expressed as $y=q^{T}k+m$, where q represents the weight coefficients.

The given expression represents the likelihood function,
(38)pCβ,  q=β2πm2exp⁡−β2 MTq−y2

In the above equation, y is the target values for regression, M is a matrix made by combining the training feature vectors horizontally, and C is the combination of $\left\{M,\;y\right\}$. β represents the noise’s inverse variance, and *T* is the total number of samples in the training set.

### 5.8. Methods for Updating Hyperparameters in Various Classifiers

The performance of a classifier greatly depends on the values assigned to its hyperparameters [55]. To find the best hyperparameter values, different methods such as Stochastic Gradient Descent (SGD), Grid Search (GS), and Adaptive Moment Estimation Method (ADAM) can be used. This study introduces a new approach called R-Adam, which aims to enhance lung cancer classification accuracy for the Decision Tree classifier and other classifiers. While Adam is a prevalent choice for hyperparameter selection in deep learning networks, this study introduces R-Adam, an adapted version proposed for hyperparameter selection across diverse classifiers. Utilizing controlled randomness, the envisioned R-Adam algorithm aims to discover hyperparameter values in proximity to the optimal values recommended by the Adam method. The investigation assesses the classification performance using both Adam and the newly introduced R-Adam technique.

#### 5.8.1. Adam Approach

The Adam approach involves employing squared gradients and exponential moving averages. The validation of hyperparameters is achieved based on the expressions provided below [56]:(39)xt+1=xt−Lrε+P^t∗Mt^
where $x_{t}\;$ represents the previous hyperparameters, $x_{t+1}$ denotes the updated hyperparameters, $L_{r}$ signifies the learning rate, and ε is a small constant used to avoid division by zero. The constants in the Adam method are $Z_{1}$ and $\;Z_{2}$.
(40)$${\hat{M}}_{t}\;=\;\frac{m_{t}}{1\;-\;Z_{1}^{t}}$$
(41)$${\hat{P}}_{t}\;=\;\frac{p_{t}}{1\;-\;Z_{2}^{t}}$$
(42)$$m_{t}=Z_{1}*m_{t-1}+(1-Z_{1})*\frac{\partial L}{\partial x_{t}}$$
(43)$$p_{t}=Z_{2}*p_{t-1}+(1-Z_{2})*{\left(\frac{\partial L}{\partial x_{t}}\right)}^{2}$$
where $\frac{\partial L}{\partial x_{t}}\;$ signifies the derivative of the loss function with respect to *x*. Thus, the mathematical representation of the loss function is as follows:(44)$$\frac{\partial L}{\partial x_{tr}}=\frac{ER_{tr}}{x_{in}},\;if\;tr=1$$
(45)$$\frac{\partial L}{\partial x_{tr}}=\frac{ER_{tr}-ER_{tr-1}}{x_{tr}-x_{tr-1}},\;if\;tr>1$$
where *ER* stands for the error rate, *tr* indicates the current iteration and *tr* − 1 denotes the previous iteration of in the Adam approach. Algorithm 1 outlines the process of utilizing the Adam optimizer to update hyperparameters in a Decision Tree model, aiming to minimize the error rate, which serves as the loss function. In Decision Tree, the key hyperparameters include maximum depth and criterion. In Decision tree, the hyperparameters are set as maximum depth = 20 and criterion = MSE. The Adam’s approach employs specific constants in this work: $L_{r}$ = 0.001, $Z_{1}$ = 0.89, $Z_{2}$ = 0.9 and $\varepsilon =10^{-9}$. Through experimentation, the optimal number of iterations for the Adam approach was determined to be 40. This iterative process aims to uncover the lowest error rate, helping identify the best hyperparameters. Notably, similar approaches involving SVM, KNN, Random Forest, SDC, MLP, and BLDC models could also leverage the Adam optimizer to update hyperparameters in a comparable manner.
**Algorithm 1. Adam Approach**Initialization: Set initial values for hyperparameters:
  Target value, maximum iterations, maximum depth and criterion for Decision Tree, maximum iterations for Adam, *L_r_*, *Z*_1_, *Z*_2_, *ε*.
2.Hyperparameter Tuning Loop:(a)For *t_r_* = 1 to maximum iterations for AdamCalculate maximum depth.Determine the criterion.(b)For *t* = 1 to maximum iterations for Decision TreeUpdate values for maximum depthCriterion is set to MSE.Determine the optimal values for maximum depth.end for.3.Formulate a confusion matrix and compute the error rate (ER).4.Compute the loss gradient using Equation (45).5.Establish new optimal hyperparameter values using Equation (39) through (43).end for.


#### 5.8.2. RAdam’s Approach

The Randomized Adam (RAdam) technique is tailored to enhance the precision of the Decision Tree classifier. Algorithm 2 presents a methodology for implementing the Decision Tree using the RAdam approach. RAdam amalgamates two core components: the Adam method and controlled randomization. The controlled randomization process is pivotal in elevating classification performance. Within each iteration of the Adam method, hyperparameters are updated. The Adam process, which meticulously refines hyperparameter ranges, is nested within the iterative controlled randomization. This controlled randomization strategy integrates two control parameters—solution considering rate and solution adjusting rate—to fulfill its objective. Constants for R-Adam are defined as follows: bandwidth is set at 0.0098, the maximum number of iterations for randomization is 15, solution considering rate is 0.6, and solution adjusting rate is 0.92. In Algorithm 2, randomization 1, randomization 2, randomization 5, and randomization 6 indicate random values from the range [0, 1], while randomization 3 and randomization 4 correspond to random values within [0, 0.1]. Following this iterative process, the lowest error rate is found, leading to the identification of optimal hyperparameters. Significantly, analogous methodologies that pertain to SVM, KNN, Random Forest, SDC, MLP, and BLDC models could also make use of the Adam optimizer for adjusting hyperparameters in a similar fashion.
**Algorithm 2. RAdam’s Approach**Initialization: Set initial values for hyperparameters:
  Target value, maximum iterations, maximum depth and criterion for Decision Tree, maximum iterations for Adam, *L_r_*, *Z*_1_, *Z*_2_, *ε*, solution considering rate, solution adjusting rate and bandwidth.
2.Hyperparameter Tuning Loop:(a)For *t_r_* = 1 to maximum iterations for AdamCalculate maximum depth.Determine the criterion.(b)For *t* = 1 to maximum iterations for Decision TreeUpdate values for maximum depthCriterion is set to MSE.Determine the optimal values for maximum depth.end for.3.Formulate a confusion matrix and compute the error rate (ER).4.Compute the loss gradient using Equation (45).5.Establish new optimal hyperparameter values using Equation (39) through (43).end for.6.For each iteration: current iterations for randomization = 1 to maximum iterations for randomizationIf randomization 1 < solution considering rate.$\mathrm{Set}\;r_{1}$$\;\mathrm{for}\;\mathrm{this}\;\mathrm{iteration}\;\mathrm{as}\;r_{1}\prime$$\mathrm{Set}\;r_{2}$$\;\mathrm{for}\;\mathrm{this}\;\mathrm{iteration}\;\mathrm{as}\;r_{2}\prime$If randomization 2 < solution adjusting rate.$\mathrm{Set}\;r_{1}$$\;\mathrm{for}\;\mathrm{this}\;\mathrm{iteration}\;\mathrm{as}\;r_{1}\prime$+bandwidth× randomization 3.$\mathrm{Set}\;r_{2}$$\;\mathrm{for}\;\mathrm{this}\;\mathrm{iteration}\;\mathrm{as}=r_{2}\prime$+bandwidth× randomization 4.end if.If $r_{1}$ for this iteration is less than the lower bound, set it to the lower bound.end if.If $r_{2}$ for this iteration is less than the lower bound, set it to the lower bound.end if.If $r_{1}$ for this iteration is less than the upper bound, set it to the upper bound.end if.If $r_{2}$ for this iteration is less than the upper bound, set it to the upper bound.end if.Set $r_{1}$ for this iteration as lower bound + (bandwidth × randomization 5).Set $r_{2}$ for this iteration as lower bound + (bandwidth × randomization 6).end if.7.Repeat.Calculate maximum depth.Determine the criterion.8.For each iteration *t* from 1 to maximum iterations for Decision TreeUpdate the values of maximum depth.Set the criterion to MSE.Determine the optimal values for maximum depth.end for.9.Formulate a confusion matrix and compute the error rate (ER).10.Compute the ER using $r_{1}$ & $r_{2}$ as hyperparameters.end for.


The following section pertains to the outcomes derived from employing diverse classification techniques for the categorization of lung cancer images within histopathological images.

## 6. Results and Discussion

This section explores the efficacy of different classifiers based on their benchmark parameters. A higher classification accuracy combined with a decreased error rate signifies robust performance of the classifier. As a result, the classifiers underwent training and testing using the extracted and chosen feature values within the Lung Histopathological Image Dataset.

### 6.1. Training and Testing of the Classifiers

The training and testing of the classifiers constitute crucial phases within classification procedures. Training facilitates the acquisition of patterns linked to the provided dimensionally reduced intensity values of the histopathological images by the classifier. In this study, the entire dataset, comprising histopathological image values related to lung cancer detection and classification, is divided into 10 equal folds. The analysis involves a series of iterations. During each iteration, one-fold is designated as the testing set, while the remaining nine folds are combined and used as the training set. In essence, 10% of the data is reserved for testing in each iteration, and the remaining 90% is utilized for training. Various performance metrics are computed for each iteration. The results obtained from all 10 iterations are collected and aggregated. This aggregation often involves calculating average values for the performance metrics. The conclusion of training and testing for the classifiers was established based on the mean square error (MSE) acting as the termination criterion. The mathematical expression for MSE is given below:(46)$$MSE=\frac{1}{M}\sum_{i=1}^{M}{\left(O_{i}-T_{k}\right)}^{2}$$
where $O_{i}$ signifies the value observed at a definite time; $T_{k}$ indicates the target value for model *k*, with “*k*” ranging from 1 to 15; and the value of *M* is assumed to be 5000 and indicates the total number of images.

### 6.2. Selection of the Optimal Parameters for the Classifiers

In this study, seven classifiers were used to categorize images into benign or adenocarcinoma based on the target selection. The target selection for the benign case $\left(T_{benign}\right)$ is represented as follows:(47)$$\frac{1}{M}\sum_{k=1}^{M}\mu_{k}\leq T_{benign}$$

The characteristics of the entire set of benign lung data (M) were subjected to normalization, and their average is denoted as $\mu_{k}$ as outlined in Equation (38), applicable for classification purposes.

The average of the normalized features is denoted as $\mu_{k}$. For benign images, a target value of 0.1 was selected, which falls within the lower end of the 0–1 scale.

The condition for choosing a target in a case of adenocarcinoma (aca) is:(48)$$\frac{1}{N}\sum_{i=1}^{N}\mu_{i}\leq T_{aca}$$

The characteristics of the entire set of lung adenocarcinoma data (N) were subjected to normalization, and their average is denoted as $\mu_{i}$ as outlined in Equation (39), applicable for classification purposes.

To enhance adenocarcinoma classification, the target selection should exceed the mean value $\mu_{k}$, which represents the average of normalized features across N images. Improving classification requires a target value of 0.5 or higher, as specified by the condition:(49)$$\left(\left\|T_{aca}-T_{benign}\right\|\;\geq 0.5\right)$$

Depending on the criteria described in Equation (40), the selected targets for this study were set at 0.1 for benign cases and 0.85 for adenocarcinoma cases. The classifiers underwent training using a 10-fold cross validation training and testing approach, with the stopping criterion being an MSE value of $\left(10^{-5}\right)$ or a maximum operation of 1000, whichever was achieved first. The selection of optimal parameters for the classifiers during the training process is outlined in Table 2. In the case of SVM (RBF) classifier the parameters are selected through the trial-and-error method. The classifiers parameters are α, Kernel width parameter (σ), w, and b are selected with the constraint of minimum MSE. In the case of KNN, K value indicates the number of clusters and in this case, it is K = 5 is selected randomly. With Euclidean Distance measure as the cluster coefficient with weight w = 0.5 is selected with the constrain of minimum MSE. In the case of Random Forest, the parameters such as Number of trees, Maximum depth and Bootstrap sample are initialized with random selection. Similarly in the case of Decision Tree, the parameter maximum depth is initialized with random selection. Since it is a binary classification problem, the class weight value for Random Forest is settled at 0.45, whereas for Decision tree, the class weight is settled at 0.4. In the case of Softmax Discriminant Classifier, it is a binary classification problem, so the λ value is settled at 0.5 along with the mean of each class target values as 0.1 and 0.85. In the case of Multilayer Perceptron Classifier, the network is trained using LM (Levenberg-Marquardt) algorithm to minimize the square output error. This error back propagation algorithm is used to calculate the weights updates in each layer of the network. As the number of hidden units gradually increased from its initial value, then there will be a reduction in the minimum Mean Squared Error (MSE) on the testing set. The optimal number of hidden units is the one that results in the lowest MSE. If the number of hidden units is increased beyond this point, the model’s performance does not show any further improvement; instead, it often starts to decline. This decline occurs since the neural network becomes unnecessarily complex, exceeding the complexity necessary to solve the problem effectively. The choice of the learning rate as 0.3 is determined based on the distribution of training patterns and their associated MSE. In case of BLDC, the parameters such as prior probability p(x) − 0.5, Class mean ${µ}_{x}$ = 0.8 and ${µ}_{y}$ = 0.1 are selected with constrain of minimum MSE. The training process demonstrated that the MSE value was attained either as low as $1.0\times 10^{-10}$ or after 1000 iterations.

### 6.3. Performance Metrics of the Classifiers

The primary objective of the classifier was to effectively distinguish between cancer cells and normal data samples in the dataset. As this research focuses on binary classification, it is essential to select appropriate performance metrics. In binary classification tasks, one of the key evaluation tools is the confusion matrix. This matrix provides a concise summary of the model’s predictions in relation to the actual labels of the dataset. The confusion matrix consists of four elements: True Positive (TP), True Negative (TN), False Positive (FP), and False Negative (FN). TP indicates the presence of lung cancer, while TN indicates its absence, both representing correct classification. FP and FN represent misclassification, where lung cancer is incorrectly predicted as present (FP), or lung cancer is present but wrongly classified as not present (FN). 

Table 3 displays TP, TN, FP, FN values, and average MSE for PSO and GWO features along with seven classifiers without employing Feature Selection Methods. Achieving the lowest MSE serves as an indicator for improved classifier performance, while a higher MSE value results in inferior classifier performance, regardless of the employed feature selection methods. PSO features show Decision Tree Classifier with the lowest MSE (3.60 × 10^−7^) and Random Forest Classifier with the highest MSE (1.60 × 10^−5^). GWO features show Bayesian LDC Classifier with the minimum MSE (2.50 × 10^−7^) and KNN Classifier with the maximum MSE (1.44 × 10^−5^).

The features extracted were given to seven classifiers for performance analysis, following feature selection methods. Table 4 shows the average MSE and confusion matrix for PSO Feature Extraction with KL Divergence and IWO feature selection. The Decision Tree had the lowest MSE (9.00 × 10^−6^) using PSO with KL Divergence, while the Bayesian LDC had the highest MSE (1.02 × 10^−5^). With PSO and IWO, the Decision Tree had the lowest MSE (7.84 × 10^−6^), while the Softmax Discriminant had the highest MSE (1.22 × 10^−5^).

Table 5 displays the average MSE and confusion matrix for GWO Feature Extraction with KL Divergence and IWO feature selection methods. The results include SVM, KNN, Random Forest, Decision Tree, Softmax Discriminant, Multilayer Perceptron, and Bayesian LDC classifiers. In the GWO with KL Divergence approach, Bayesian LDC achieves the lowest MSE (1.00 × 10^−8^), while the Multilayer Perceptron Classifier has the highest MSE (2.03 × 10^−5^). Similarly, in the PSO with IWO approach, SVM achieves the minimum MSE (4.90 × 10^−7^), while the Random Forest Classifier has the maximum MSE (1.52 × 10^−5^).

Table 6 presents the mean Mean Squared Error (MSE) and confusion matrix outcomes for PSO Feature Extraction using KL Divergence and IWO feature selection techniques in Adam Hyperparameter Tuning. Among these, Bayesian LDC achieved the smallest MSE (8.41 × 10^−6^) through PSO with KL Divergence, whereas Random Forest showed the highest MSE (2.72 × 10^−4^). When considering PSO and IWO, Random Forest demonstrated the lowest MSE (9.00 × 10^−8^), whereas Softmax Discriminant had the highest MSE (4.00 × 10^−4^).

Table 7 displays the average Mean Squared Error (MSE) and the results of the confusion matrices obtained from GWO Feature Extraction using KL Divergence and IWO feature selection techniques in Adam Hyperparameter Tuning. Among these approaches, Multilayer Perceptron achieved the smallest MSE of 6.40 × 10^−7^ using GWO with KL Divergence, while SVM exhibited the highest MSE of 1.23 × 10^−5^. Considering both GWO and IWO, Decision Tree showcased the lowest MSE of 6.40 × 10^−7^, whereas Softmax Discriminant had the highest MSE of 1.04 × 10^−4^.

Table 8 presents the average Mean Squared Error (MSE) and the results of confusion matrices obtained by using PSO Feature Extraction with KL Divergence and IWO feature selection techniques during R-Adam Hyperparameter Tuning. Among these methods, SVM achieved the smallest MSE of 6.56 × 10^−5^ when using GWO with KL Divergence, while Random Forest had the highest MSE of 1.09 × 10^−5^. Considering both PSO and IWO, Random Forest had the lowest MSE of 4.49 × 10^−5^, while Softmax Discriminant had the highest MSE of 1.10 × 10^−4^.

Table 9 displays the average Mean Squared Error (MSE) and the outcomes of confusion matrices. These were derived using GWO Feature Extraction with KL Divergence and IWO feature selection methods within R-Adam Hyperparameter Tuning. Among the techniques, Bayesian LDC achieved the lowest MSE of 9.61 × 10^−6^ with GWO and KL Divergence. Conversely, Random Forest had the highest MSE of 1.02 × 10^−5^. When considering both GWO and IWO, KNN displayed the smallest MSE of 5.48 × 10^−5^, while Random Forest exhibited the highest MSE of 1.90 × 10^−4^.

Table 10 presents the metrics used to evaluate the performance of classifiers, including Accuracy, Error Rate, F1 Score, MCC, Jaccard Index, g-Mean, and Kappa. The mathematical expressions for these metrics are also provided.

The lung cancer data are processed using PSO and GWO techniques to extract features from normal and malignant data. These features are then used as inputs for seven classification models. Table 11 shows the performance of the classifiers without Feature Selection. The Decision Tree Classifier stands out with the highest accuracy of 85.01% for GWO features. It also achieves the highest F1 score (85.77%), MCC value (0.70), Jaccard Index (75.08%), g-mean (85.33%), kappa score (0.70), and the lowest error rate (14.99%). In contrast, the Random Forest classifier performs poorly for PSO features, with an accuracy of 56.25%, F1 score of 55.17%, MCC value of 0.13, Jaccard Index of 38.09%, g-mean of 56.26%, kappa value of 0.13, and the highest error rate of 43.75%. Without feature selection, the Decision Tree Classifier with GWO feature extraction method achieves the best accuracy and outperforms other classifiers.

The performance of a random forest model heavily relies on the quality of individual trees and the diversity among them. If a random forest includes subpar or correlated trees, it can result in reduced overall accuracy. Correlation among trees can introduce redundant information, hampering the model’s ability to generalize effectively to new data and causing a drop in accuracy.

In Table 11, the Random Forest Classifier, when paired with the feature extraction technique of Particle Swarm Optimization (PSO), achieves a lower accuracy of 56.25% compared to the Genetic Wolf Optimization (GWO) approach, which achieves an accuracy of 77.84%. This discrepancy is primarily due to PSO selecting suboptimal intensity values of the segmented image, including less informative or irrelevant intensity values in the random forest model. Effective selection of intensity values is crucial for any classifier to yield better results. If the process of selecting intensity values fails to filter out irrelevant ones, it can negatively impact the performance of the Random Forest model. And also, PSO selects intensity values that are highly specific to the training dataset, resulting in a model that performs well on the training data but struggles to generalize to new, unseen data. Furthermore, the computational demands of PSO can lead to longer tree construction times, affecting the overall classifier’s performance.

Table 12 compares the performance of seven classifiers with PSO, KL Divergence, and IWO Feature Selection. The Softmax Discriminant Classifier stands out with superior results for KL Divergence features, achieving an accuracy of 83.47%, the highest F1 score of 83.18%, MCC of 0.67, Jaccard Index of 71.21%, g-mean of 83.50%, Kappa score of 0.67, and the lowest error rate of 16.53%. Conversely, the Bayesian LDC classifier performs poorly with IWO features, obtaining an accuracy of 53.79%, F1 score of 53.71%, MCC of 0.08, Jaccard Index of 36.72%, g-mean of 53.79%, Kappa of 0.08, and the highest error rate of 46.21%. Overall, the Softmax Discriminant Classifier using PSO and KL Divergence Feature selection achieves the highest accuracy and outperforms other classifiers.

Table 13 presents the performance of seven classifiers using GWO features, KL Divergence, and IWO Feature Selection. The KNN Classifier achieves the highest accuracy of 79.36% with KL Divergence features. It also obtains the highest F1 score (78.60%), MCC value (0.59), Jaccard Index (64.74%), g-mean (79.49%), kappa score (0.59), and lowest error rate (20.64%) among all classifiers. However, the Bayesian LDC classifier performs poorly with IWO features, achieving an accuracy of 59.76%, F1 score of 61.52%, MCC value of 0.20, Jaccard Index of 44.42%, g-mean of 59.84%, kappa value of 0.20, and the highest error rate (40.24%). The KNN Classifier with GWO and KL Divergence Feature selection method demonstrates the best accuracy and outperforms other classifiers.

Table 14 presents a comprehensive performance analysis of various classifiers, utilizing PSO with KL Divergence and IWO in combination with Adam Hyperparameter Tuning. The findings highlight that the K-Nearest Neighbors (KNN) Classifier attains the highest accuracy at 86.70% when incorporating KL Divergence features. This classifier also excels in other evaluation metrics, boasting the highest F1 score (86.30%), MCC value (0.74), Jaccard Index (75.96%), geometric mean (g-mean) (86.81%), kappa score (0.73), and displaying the lowest error rate (13.30%) compared to all other classifiers. Conversely, the performance of the Bayesian Linear Discriminant Classifier (LDC) is notably subpar when employing IWO features, achieving an accuracy of 76.14%, an F1 score of 76.19%, an MCC value of 0.52, a Jaccard Index of 61.54%, a g-mean of 76.14%, a kappa value of 0.52, and the highest error rate (23.86%) among the classifiers considered. Overall, the KNN Classifier in conjunction with PSO and the KL Divergence Feature selection method emerges as the standout performer, showcasing superior accuracy and outclassing the other classifiers in the evaluation.

Table 15 provides a comprehensive analysis of classifier performance, utilizing a combination of PSO with KL Divergence and IWO along with R-Adam Hyperparameter Tuning. The results highlight that the K-Nearest Neighbors (KNN) Classifier achieves the highest accuracy of 87.45% when incorporating KL Divergence features. This classifier also excels across various evaluation metrics, including the highest F1 score (87.02%), MCC value (0.75), Jaccard Index (77.03%), geometric mean (g-mean) (87.58%), kappa score (0.75), and the lowest error rate (12.55%) compared to other classifiers. On the other hand, the Decision Tree’s performance is notably weaker when using KL Divergence features, with an accuracy of 78.09%, F1 score of 78.19%, MCC value of 0.55, Jaccard Index of 64.19%, g-mean of 78.10%, kappa value of 0.54, and the highest error rate (21.91%) among considered classifiers. In summary, the KNN Classifier, in combination with PSO and the KL Divergence Feature selection method, stands out as the top performer, showcasing exceptional accuracy and surpassing other classifiers in the evaluation.

Table 16 provides a comprehensive analysis of classifier performance, utilizing GWO with KL Divergence and IWO along with Adam Hyperparameter Tuning. The results emphasize that the K-Nearest Neighbors (KNN) Classifier achieves the highest accuracy at 90.87% when incorporating KL Divergence features. This classifier also excels in various evaluation metrics, recording the highest F1 score (90.06%), MCC value (0.83), Jaccard Index (81.92%), geometric mean (g-mean) (91.71%), kappa score (0.82), and demonstrating the lowest error rate (9.14%) compared to alternative classifiers. In contrast, the performance of the Decision Tree is notably below par when utilizing KL Divergence features, attaining an accuracy of 76.20%, an F1 score of 75.33%, an MCC value of 0.53, a Jaccard Index of 60.43%, a g-mean of 76.30%, a kappa score of 0.52, and the highest error rate (23.81%) among the considered classifiers. In summary, the KNN Classifier, combined with GWO and the KL Divergence Feature selection approach, stands out as the top performer, showcasing remarkable accuracy and surpassing the other classifiers in the evaluation.

Table 17 presents a comprehensive analysis of classifier performance, utilizing GWO with KL Divergence, and IWO alongside R-Adam Hyperparameter Tuning. The outcomes underscore the Decision Tree Classifier’s exceptional performance, achieving the highest accuracy at 91.57% when integrating KL Divergence features. This classifier also outperforms others across various evaluation metrics, achieving the highest F1 score (91.71%), MCC value (0.83), Jaccard Index (84.70%), geometric mean (g-mean) (91.87%), kappa score (0.83), and demonstrating the lowest error rate (8.43%) compared to alternative classifiers. In contrast, the performance of the Decision Tree notably drops when utilizing IWO, with an accuracy of 77.35%, an F1 score of 76.80%, an MCC value of 0.55, a Jaccard Index of 62.34%, a g-mean of 77.39%, a kappa score of 0.55, and the highest error rate (22.66%) among the considered classifiers. In recap, the Decision Tree classifier, combined with GWO and the KL Divergence Feature selection approach, emerges as the leading performer, showcasing remarkable accuracy and surpassing the other classifiers in the evaluation.

Table 18 presents a summary of the performance outcomes for each combination of feature extraction and feature selection using Adam and R-Adam Hyperparameter tuning methods across all seven classifiers. The highest accuracy of 91.57% in the Decision Tree classifier was attained by combining GWO and IWO techniques, utilizing the RAdam Hyperparameter tuning approach.

Figure 7 illustrates the comparative performance of classifiers in relation to Accuracy, both with and without the integration of feature selection. As depicted in the graph, among all of the classifier types, the Decision Tree classifier employing the GWO (Grey Wolf Optimization) feature extraction method outperformed the rest in terms of achieving the highest accuracy. When utilizing the KL Divergence feature selection technique, along with PSO feature extraction technique, the Softmax Discriminant Classifier demonstrated a commendable accuracy of 83.47%. Similarly, when employing the IWO (Invasive Weed Optimization) feature selection technique, along with GWO feature extraction technique, the SVM (Support Vector Machine) classifier exhibited a notable accuracy of 76.63%. In contrast, the Mathematical feature selection approaches yielded comparatively lower accuracy when compared to scenarios where feature selection was not applied.

Figure 8 displays how classifiers perform when hyperparameter tuning methods such as Adam and RAdam are used to enhance accuracy. Even after applying feature selection techniques, there’s no significant improvement in classifier accuracy compared to using no feature selection. To address this, hyperparameter update algorithms are introduced. The accuracy achieved through the KL Divergence feature selection method is notably high across all classifiers. However, for the IWO feature selection technique, accuracy seems to be somewhat lower. This prompts the use of hyperparameter update algorithms specifically for the IWO feature selection. As a result of employing these algorithms, there’s a substantial accuracy improvement for all classifiers using the IWO feature selection. Notably, the Decision Tree classifier combined with GWO feature extraction and IWO feature selection, along with the RAdam hyperparameter update algorithm, achieves the highest accuracy at 91.57%.

Figure 9 illustrates the classifiers’ performance by analyzing the Deviation of MCC and Kappa parameters in relation to their mean values. These parameters, MCC and Kappa, serve as benchmarks for evaluating how classifiers respond to different inputs. The study involves two input categories: features extracted using PSO and GWO, followed by feature selection through KL Divergence and IWO. The selected features are then inputted into the classifiers, and their effectiveness is evaluated through the resulting MCC and Kappa values. The average MCC and Kappa values attained from the classifiers are 0.56661 and 0.56256, respectively. A methodology is devised to assess classifier performance by examining the variability of MCC and Kappa values from their respective means. Notably, Figure 9 depicts a trend where MCC and Kappa values in the graph’s third quadrant correspond to non-linear outcomes with lower performance metrics. Conversely, values in the graph’s first quadrant indicate improved classifier performance, with MCC and Kappa values surpassing the average. This pattern suggests an enhancement in classifier performance for GWO inputs when coupled with IWO feature selection, particularly within the context of the RAdam hyperparameter tuning approach. Figure 9 is also characterized by a linear curve fitting described by the equation Y = 1.0017 X + 4 × 10^−06^, with an R^2^ value of 0.998.

### 6.4. Computational Complexity Analysis of the Classifiers

Computational complexity also acts as a performance metric for classifiers, encompassing time and space complexities. This study utilizes the Big O notation to characterize the computational complexity of feature extraction, feature selection, and classification methods. The assessment of Computational Complexity involves an input size labelled as ‘*n*’. When the input size is O(1), the computational complexity remains minimal. However, as the input size increases, so does the computational complexity. The relationship between input size and computational complexity is encapsulated by the Big O notation. Specifically, if the complexity grows logarithmically with the increase in ‘n’, it is represented as O(log⁡n). The classifiers examined in this research integrate either feature extraction methods, feature selection techniques, or a combination of both. Hence, computational complexity becomes a blend of these hybrid methodologies. Table 19 offers an overview of the computational complexity associated with the classifiers across diverse Feature Extraction and Feature Selection Techniques.

As evident from Table 19, when feature extraction techniques are not employed, classifiers such as SVM, KNN, Random Forest (RF), Decision Tree (DT), Softmax Discriminant classifier (SDC), Multilayer Perceptron (MLP), and Bayesian LDC (BLDC) exhibit lower levels of computational complexity. When utilizing the GWO feature extraction technique, the Decision Tree classifier stands out with a computational complexity of $O\;(n^{3}\log n)$ and achieves a high accuracy of 85.01%. However, the BLDC classifier, with a computational complexity of $O\;(n^{6}\log n)$ for GWO feature extraction, performs poorly when IWO feature selection methods are applied across the classifiers. The observed underperformance is linked to outlier problems present in the GWO features. To improve classifier performance, this study integrates hyperparameter tuning into the IWO feature selection method. Particularly, the Decision Tree classifier demonstrates remarkable performance with accuracies of 90.07% and 91.57% when utilizing GWO feature extraction in conjunction with IWO. These improvements come with a moderate computational complexity represented by $O({2n}^{5}\log3n)$ for the Adam Hyperparameter tuning approach and $O({2n}^{5}\log6n)$ for the RAdam Hyperparameter tuning approach.

### 6.5. Comparison of Previous Works

Comparison charts with different Machine Learning and Deep Learning models along with classifiers are shown in the Table 20 for the different datasets of lung cancer. As noted in the Table 20, that the classifier performance is analyzed for the four different datasets, namely CRAG, LIDC-IDRI, LUNA16 and LC25000. The following classifiers namely Ensemble, ResNet50, KNN, AlexNet and CNN were analyzed. As shown in the Table 20., for the CRAG database, the ResNet50 attains the maximum accuracy of 93.91% whereas for CT image database (LIDC-IDRI, LUNA16), the CNN-ALCDC model attains the maximum accuracy of 97.2%. This is due to the smaller number of CT images as the data set. Similarly, for LC25000 database, for a binary classification problem, 5000 images for each class are taken, which attains a commendable accuracy of 91.57%.

## 7. Conclusions

Early diagnosis of lung cancer enhances patient life expectancy. This paper proposes machine learning techniques to enhance classifier accuracy and enable early identification using histopathological images. The primary aim is to achieve lung cancer classification with high accuracy, while minimizing false positives and false negatives. The study applies adaptive median filtering and a modified KFCM-based segmentation method to obtain the segmented images for better classification results. Feature extraction involves optimization techniques such as PSO and GWO which reduces the dimensionality of the segmented image to [512 × 10], followed by statistical analysis. Feature selection reduces the number of intensity values to [100 × 12] for lung cancer classification. Through the utilization of KL Divergence and Invasive Weed Optimization to evaluate the dimensionally reduced features, the datasets undergo classification with various classifiers to achieve better accuracy. The classification process entails seven classifiers, coupled with Hyperparameter selection using Adam and Radam methods, which are compared and analyzed. The Decision Tree Classifier for GWO features without feature selection achieves a better accuracy of 85.01%. Mathematical feature selection methods had lower accuracy compared to scenarios without feature selection. The results are further enhanced when the Hyperparameter Up-to-date methods are employed which reveal that the combination of GWO-IWO-Decision Tree classifier for RAdam outperforms all other classifiers, achieving an overall accuracy of 91.57% in classifying Benign and Adenocarcinoma classes. Future research directions will explore diverse feature selection techniques, optimization methodologies, and the inclusion of deep learning approaches such as CNN, DNN, and LSTM to further enhance lung cancer classification.

## Figures and Tables

**Figure 1 diagnostics-13-03289-f001:**
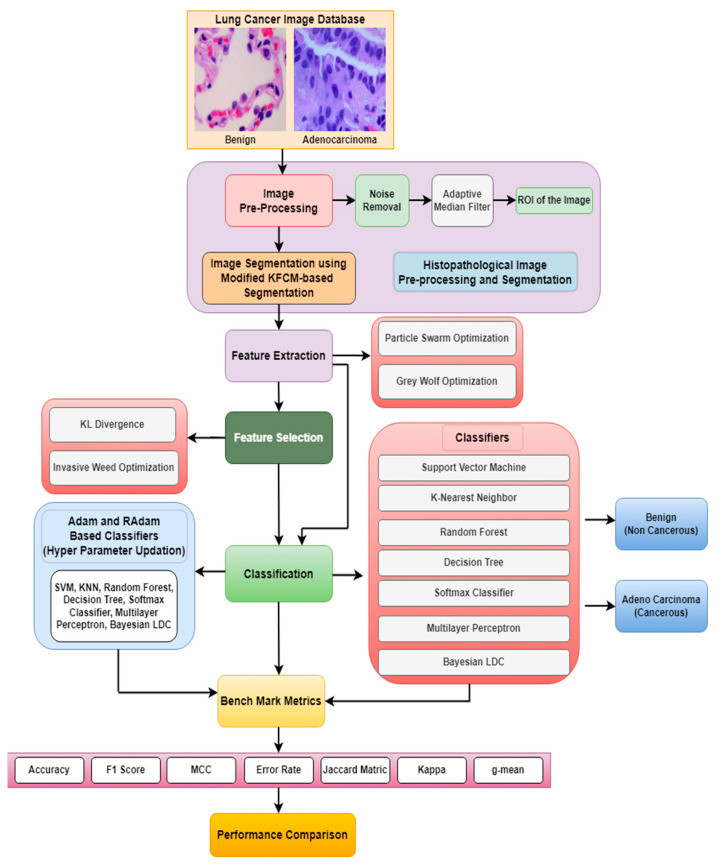
Schematic representation for detecting lung abnormalities from Histopathological Images.

**Figure 2 diagnostics-13-03289-f002:**
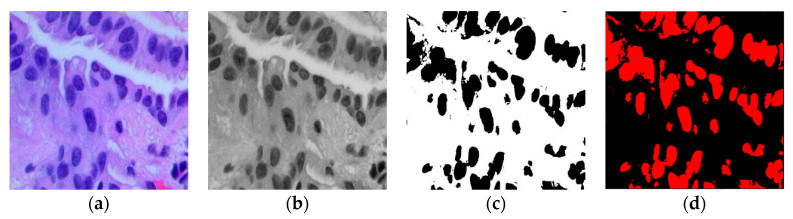
(**a**) Original ACA image; (**b**) Filtered ACA image; (**c**) ROI of the ACA Image; (**d**) Segmented ACA image.

**Figure 3 diagnostics-13-03289-f003:**
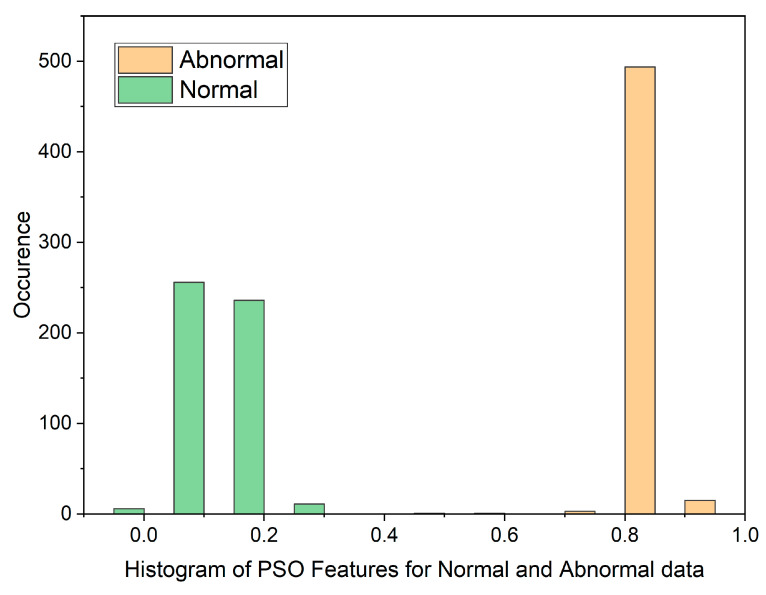
Histogram of PSO Features for Normal and Malignant data.

**Figure 4 diagnostics-13-03289-f004:**
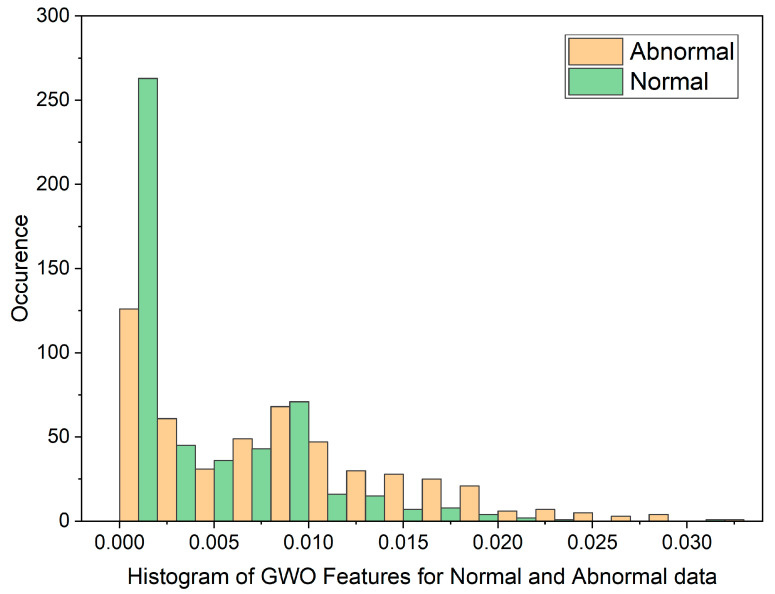
Histogram of GWO Features for Normal and Malignant data.

**Figure 5 diagnostics-13-03289-f005:**
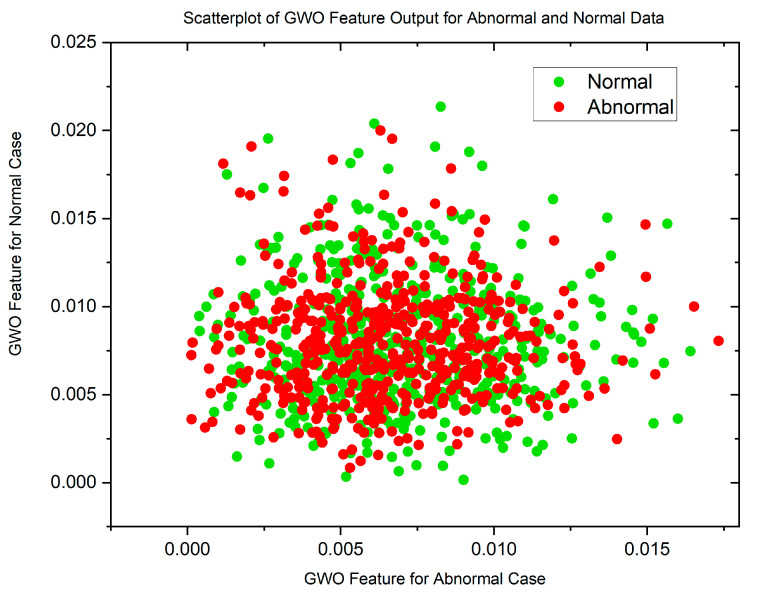
Scatterplot of PSO Features for Normal and Malignant Case.

**Figure 6 diagnostics-13-03289-f006:**
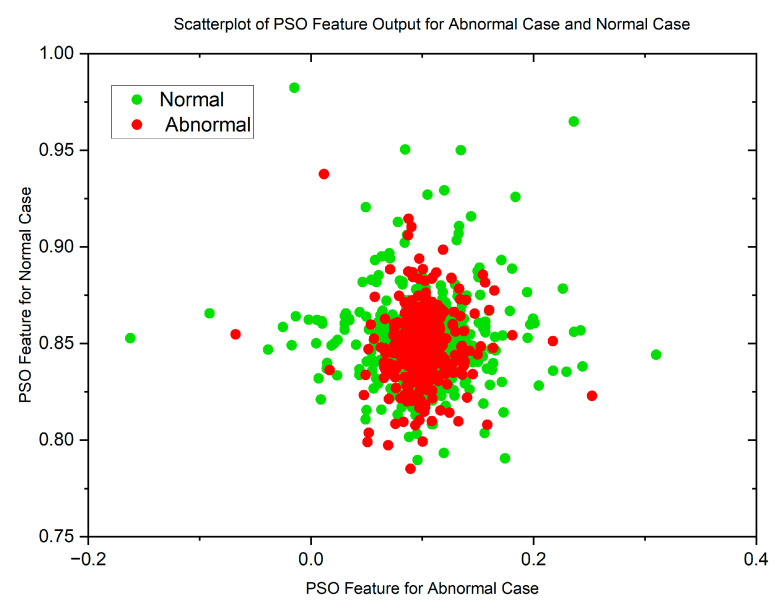
Scatterplot of GWO Features for Normal and Malignant Case.

**Figure 7 diagnostics-13-03289-f007:**
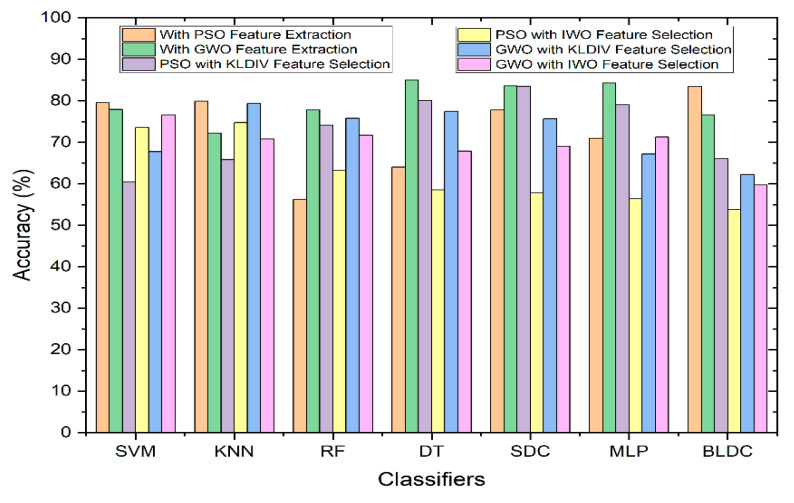
Performance of Classifiers with and without Feature Selection Methods in terms of Accuracy.

**Figure 8 diagnostics-13-03289-f008:**
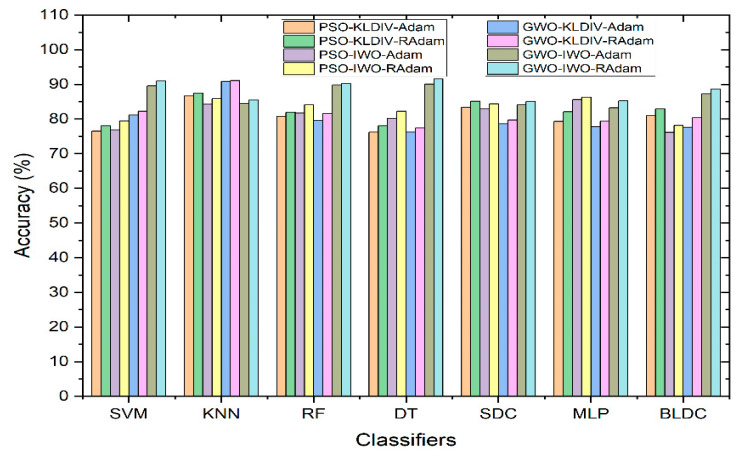
Performance of Classifiers with Hyperparameter Tuning Methods for Adam and RAdam in terms of Accuracy.

**Figure 9 diagnostics-13-03289-f009:**
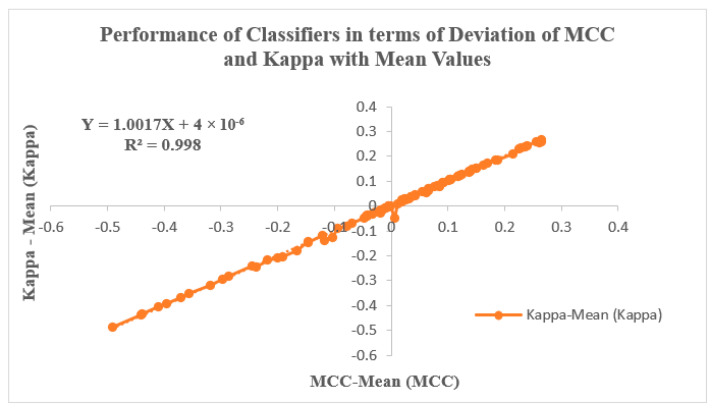
Performance of Classifiers in terms of Deviation of MCC and Kappa Parameters with mean values.

**Table 1 diagnostics-13-03289-t001:** Statistical Parameters in PSO and GWO for Feature Extraction in Malignant and Normal Data.

Statistical Parameters	PSO		GWO	
	Malignant	Normal	Malignant	Normal
Mean	0.8598080214	0.1109701363	0.01878313748	0.01751341349
Variance	0.05867975074	0.07425036326	0.07492946326	0.07494543857
Skewness	19.87029488	19.83047771	22.52231557	22.56212107
Kurtosis	441.8828416	444.9961882	509.1565306	510.3537192
Pearson CC	0.9019022281	0.9269991469	0.9985202125	0.997858273
CCA	0.12309		0.11291	

**Table 2 diagnostics-13-03289-t002:** Selection of the Optimal Parameters for the Classifiers.

Classifiers	Optimal Parameters of the Classifiers
Support Vector Machine	Kernel—RBF; α—1; Kernel width parameter (σ)—100; w—0.85; b—0.01; Convergence Criterion—MSE.
K-Nearest Neighbor	K—5; Distance Metric—Euclidian; w—0.5; Criterion—MSE.
Random Forest	Number of Trees—200; Maximum Depth—10; Bootstrap Sample—20; Class Weight—0.45.
Decision Tree	Maximum Depth—20; Impurity Criterion—MSE; Class Weight—0.4.
Softmax Discriminant Classifier	λ = 0.5 along with mean of each class target values as 0.1 and 0.85.
Multilayer Perceptron	Learning rate—0.3; Learning Algorithm—LM; Criterion—MSE.
Bayesian Linear Discriminant Classifier	Prior Probability P(x)—0.5; Class mean ${µ}_{x}$ = 0.8 and ${µ}_{y}$ = 0.1, Criterion = MSE.

**Table 3 diagnostics-13-03289-t003:** Confusion Matrix for Classifiers without Feature Selection.

Feature Extraction	Classifiers	Confusion Matrix				MSE
		TP	TN	FP	FN	
PSO	SVM	3944	4009	991	1056	7.29 × 10^−6^
	KNN	4267	3725	1275	733	4.49 × 10^−5^
	Random Forest	2692	2933	2067	2308	1.60 × 10^−5^
	Decision Tree	3184	3217	1783	1816	3.60 × 10^−7^
	Softmax Discriminant	4033	3750	1250	967	4.00 × 10^−8^
	Multilayer Perceptron	3425	3675	1325	1575	2.25 × 10^−6^
	Bayesian LDC	4367	3975	1025	633	5.63 × 10^−5^
GWO	SVM	3617	4175	825	1383	5.76 × 10^−6^
	KNN	3500	3725	1275	1500	1.44 × 10^−5^
	Random Forest	3967	3817	1183	1033	3.36 × 10^−5^
	Decision Tree	4517	3984	1016	483	8.41 × 10^−6^
	Softmax Discriminant	4083	4275	725	917	1.96 × 10^−4^
	Multilayer Perceptron	4050	4384	616	950	4.84 × 10^−4^
	Bayesian LDC	3967	3692	1308	1033	2.50 × 10^−7^

**Table 4 diagnostics-13-03289-t004:** Confusion Matrix for Classifiers for PSO with KL Divergence and IWO.

Feature Selection	Classifiers	Confusion Matrix				MSE
		TP	TN	FP	FN	
KL Divergence	SVM	3297	2747	2253	1703	3.24 × 10^−6^
	KNN	3978	2605	2395	1022	8.41 × 10^−6^
	Random Forest	4115	3294	1706	885	2.30 × 10^−5^
	Decision Tree	3919	4089	911	1081	9.00 × 10^−6^
	Softmax Discriminant	4089	4258	742	911	4.84 × 10^−6^
	Multilayer Perceptron	4271	3633	1367	729	2.56 × 10^−6^
	Bayesian LDC	3298	3311	1690	1702	1.02 × 10^−5^
IWO	SVM	3854	3503	1497	1146	2.21 × 10^−5^
	KNN	3490	3985	1016	1510	3.36 × 10^−5^
	Random Forest	3574	2757	2243	1426	1.94 × 10^−5^
	Decision Tree	2982	2871	2129	2018	7.84 × 10^−6^
	Softmax Discriminant	2734	3047	1953	2266	1.22 × 10^−5^
	Multilayer Perceptron	3047	2592	2408	1953	1.00 × 10^−6^
	Bayesian LDC	2681	2698	2302	2319	1.85 × 10^−5^

**Table 5 diagnostics-13-03289-t005:** Confusion Matrix for Classifiers for GWO with KL Divergence and IWO.

Feature Selection	Classifiers	Confusion Matrix				MSE
		TP	TN	FP	FN	
KL Divergence	SVM	4029	2742	2258	971	1.00 × 10^−6^
	KNN	3789	4147	853	1211	4.90 × 10^−5^
	Random Forest	3490	4089	911	1510	6.40 × 10^−7^
	Decision Tree	3594	4147	853	1406	2.50 × 10^−7^
	Softmax Discriminant	4896	2668	2333	104	1.00 × 10^−6^
	Multilayer Perceptron	3737	2982	2018	1263	2.03 × 10^−5^
	Bayesian LDC	3460	2767	2233	1540	1.00 × 10^−8^
IWO	SVM	4401	3262	1738	599	4.90 × 10^−7^
	KNN	3203	3880	1120	1797	1.60 × 10^−5^
	Random Forest	4440	2735	2265	560	1.52 × 10^−5^
	Decision Tree	4167	2620	2380	833	5.29 × 10^−6^
	Softmax Discriminant	4219	2687	2313	781	2.30 × 10^−5^
	Multilayer Perceptron	4375	2747	2253	625	9.61 × 10^−6^
	Bayesian LDC	3216	2760	2240	1784	6.89 × 10^−5^

**Table 6 diagnostics-13-03289-t006:** Confusion Matrix for Classifiers: PSO with KL Divergence and IWO for Adam Hyperparameter Tuning.

Feature Selection	Classifiers	Confusion Matrix				MSE
		TP	TN	FP	FN	
KL Divergence	SVM	4089	3568	1433	911	6.61 × 10^−4^
	KNN	4184	4487	514	817	1.44 × 10^−5^
	Random Forest	4555	3520	1480	445	2.72 × 10^−4^
	Decision Tree	3815	3809	1191	1185	6.72 × 10^−5^
	Softmax Discriminant	4392	3948	1052	608	2.40 × 10^−5^
	Multilayer Perceptron	3881	4048	952	1119	1.96 × 10^−6^
	Bayesian LDC	4156	3947	1053	844	8.41 × 10^−6^
IWO	SVM	3599	4085	915	1401	8.10 × 10^−5^
	KNN	4058	4375	625	942	7.23 × 10^−5^
	Random Forest	4129	4038	962	871	9.00 × 10^−8^
	Decision Tree	3713	4308	692	1288	6.40 × 10^−5^
	Softmax Discriminant	4129	4161	839	871	4.00 × 10^−4^
	Multilayer Perceptron	4539	4024	976	461	2.50 × 10^−5^
	Bayesian LDC	3817	3797	1203	1183	1.44 × 10^−5^

**Table 7 diagnostics-13-03289-t007:** Confusion Matrix for Classifiers: GWO with KL Divergence and IWO for Adam Hyperparameter Tuning.

Feature Selection	Classifiers	Confusion Matrix				MSE
		TP	TN	FP	FN	
KL Divergence	SVM	3653	4466	534	1347	1.23 × 10^−5^
	KNN	4139	4948	52	862	7.23 × 10^−5^
	Random Forest	4044	3913	1088	956	1.30 × 10^−5^
	Decision Tree	3635	3985	1016	1365	6.89 × 10^−5^
	Softmax Discriminant	3565	4297	703	1435	1.37 × 10^−5^
	Multilayer Perceptron	3740	4034	966	1260	6.40 × 10^−7^
	Bayesian LDC	3775	3987	1013	1225	4.90 × 10^−7^
IWO	SVM	4339	4617	383	661	1.94 × 10^−5^
	KNN	4129	4321	680	871	5.76 × 10^−6^
	Random Forest	4509	4466	534	491	7.57 × 10^−5^
	Decision Tree	4617	4390	610	383	6.40 × 10^−7^
	Softmax Discriminant	4409	4005	995	592	1.04 × 10^−4^
	Multilayer Perceptron	4409	3913	1088	592	4.49 × 10^−5^
	Bayesian LDC	4754	3973	1027	246	4.90 × 10^−7^

**Table 8 diagnostics-13-03289-t008:** Confusion Matrix for Classifiers: PSO with KL Divergence and IWO for RAdam Hyperparameter Tuning.

Feature Selection	Classifiers	Confusion Matrix				MSE
		TP	TN	FP	FN	
KL Divergence	SVM	4144	3668	1333	856	6.56 × 10^−5^
	KNN	4209	4537	464	792	2.92 × 10^−5^
	Random Forest	4575	3620	1380	425	1.09 × 10^−5^
	Decision Tree	3950	3859	1141	1050	5.93 × 10^−5^
	Softmax Discriminant	4417	4098	902	583	1.60 × 10^−5^
	Multilayer Perceptron	4011	4198	802	989	3.03 × 10^−5^
	Bayesian LDC	4245	4047	953	755	3.97 × 10^−5^
IWO	SVM	3710	4235	765	1290	1.37 × 10^−5^
	KNN	4208	4375	625	792	4.22 × 10^−5^
	Random Forest	4229	4188	812	771	4.49 × 10^−5^
	Decision Tree	3813	4408	592	1188	4.36 × 10^−5^
	Softmax Discriminant	4229	4211	789	771	1.10 × 10^−4^
	Multilayer Perceptron	4558	4074	926	443	2.30 × 10^−5^
	Bayesian LDC	3917	3897	1103	1083	3.02 × 10^−5^

**Table 9 diagnostics-13-03289-t009:** Confusion Matrix for Classifiers: GWO with KL Divergence and IWO for RAdam Hyperparameter Tuning.

Feature Selection	Classifiers	Confusion Matrix				MSE
		TP	TN	FP	FN	
KL Divergence	SVM	3758	4466	534	1242	1.37 × 10^−5^
	KNN	4159	4948	52	842	2.40 × 10^−5^
	Random Forest	4094	4063	938	906	1.02 × 10^−5^
	Decision Tree	3750	3985	1016	1250	1.23 × 10^−5^
	Softmax Discriminant	3670	4297	703	1330	4.76 × 10^−5^
	Multilayer Perceptron	3860	4084	916	1140	2.12 × 10^−5^
	Bayesian LDC	3905	4137	863	1095	9.61 × 10^−6^
IWO	SVM	4439	4667	333	561	4.36 × 10^−5^
	KNN	4229	4321	680	771	5.48 × 10^−5^
	Random Forest	4559	4466	534	441	1.90 × 10^−4^
	Decision Tree	4667	4490	510	333	2.40 × 10^−5^
	Softmax Discriminant	4459	4055	945	542	5.33 × 10^−5^
	Multilayer Perceptron	4459	4063	938	542	5.04 × 10^−5^
	Bayesian LDC	4789	4073	927	211	1.09 × 10^−5^

**Table 10 diagnostics-13-03289-t010:** Standard Benchmark Parameters.

Performance Metrics	Equation	Significance
Accuracy (%)	$Accuracy=\frac{TP+TN}{TP+TN+FP+FN}$	Average positive-to-negative sample ratio.
Error Rate	$Err=\frac{FP+FN}{TP+TN+FP+FN}$	The number of incorrect predictions, based on recorded observations.
F1 Score (%)	$F1=\frac{2TP}{2TP+FP+FN}$	Average of precision and recall to obtain the classification accuracy of a specific class.
MCC	$MCC=\frac{TN\times TP-FN\times FP}{\sqrt{\left(TP+FP)(TP+FN)(TN+FP)(TN+FN\right)}}$	Pearson correlation between the actual output and the achieved output.
Jaccard Index (%)	$Jaccard=\frac{TP}{TP+FP+FN}$	The number of predicted true positives exceeded the number of actual positives, regardless of whether they were real or predicted.
g-mean (%)	$g-mean=\sqrt{\frac{TP}{TP+FN}*\frac{TN}{TN+FP}}$	Combination of sensitivity and specificity into a single value that balances both objectives.
Kappa	$Kappa=\frac{\mathrm{Pr}\left(a\right)-Pr(e)}{1-Pr(e)}\;$	Inter-rater agreement measure for assessing agreement between two methods in categorizing cancer cases.

**Table 11 diagnostics-13-03289-t011:** Performance Analysis of the Classifiers without Feature Selection.

Feature Extraction	Classifiers	Accuracy (%)	Error Rate (%)	F1 Score (%)	MCC	Jaccard Index (%)	g-Mean (%)	Kappa
PSO	SVM	79.53	20.47	79.40	0.59	65.83	79.53	0.59
	KNN	79.92	20.08	80.95	0.60	68.00	80.21	0.60
	Random Forest	56.25	43.75	55.17	0.13	38.09	56.26	0.13
	Decision Tree	64.01	35.99	63.89	0.28	46.94	64.01	0.28
	Softmax Discriminant	77.83	22.17	78.44	0.56	64.53	77.90	0.56
	Multilayer Perceptron	71	29	70.26	0.42	54.15	71.04	0.42
	Bayesian LDC	83.42	16.58	84.05	0.67	72.48	83.59	0.67
GWO	SVM	77.92	22.08	76.62	0.56	62.09	78.21	0.56
	KNN	72.25	27.75	71.61	0.45	55.78	72.29	0.45
	Random Forest	77.84	22.16	78.17	0.56	64.16	77.86	0.56
	Decision Tree	85.01	14.99	85.77	0.70	75.08	85.33	0.70
	Softmax Discriminant	83.58	16.42	83.26	0.67	71.32	83.62	0.67
	Multilayer Perceptron	84.34	15.66	83.80	0.69	72.12	84.46	0.69
	Bayesian LDC	76.59	23.41	77.22	0.53	62.89	76.66	0.53

**Table 12 diagnostics-13-03289-t012:** Performance Analysis of the Classifiers for PSO with KL Divergence and IWO.

Feature Selection	Classifiers	Accuracy (%)	Error Rate (%)	F1 Score (%)	MCC	Jaccard Index (%)	g-mean (%)	Kappa
KL Divergence	SVM	60.44	39.56	62.51	0.21	45.46	60.56	0.21
	KNN	65.83	34.17	69.96	0.33	53.79	66.96	0.32
	Random Forest	74.09	25.91	76.05	0.49	61.36	74.65	0.48
	Decision Tree	80.08	19.92	79.74	0.60	66.30	80.11	0.60
	Softmax Discriminant	83.47	16.53	83.18	0.67	71.21	83.50	0.67
	Multilayer Perceptron	79.04	20.96	80.30	0.59	67.08	79.43	0.58
	Bayesian LDC	66.08	33.92	66.04	0.32	49.30	66.08	0.32
IWO	SVM	73.57	26.43	74.47	0.47	59.32	73.67	0.47
	KNN	74.74	25.26	73.43	0.50	58.01	74.95	0.49
	Random Forest	63.31	36.69	66.09	0.27	49.35	63.64	0.27
	Decision Tree	58.53	41.47	58.99	0.17	41.83	58.54	0.17
	Softmax Discriminant	57.81	42.19	56.45	0.16	39.32	57.84	0.16
	Multilayer Perceptron	56.39	43.61	58.29	0.13	41.13	56.44	0.13
	Bayesian LDC	53.79	46.21	53.71	0.08	36.72	53.79	0.08

**Table 13 diagnostics-13-03289-t013:** Performance Analysis of the Classifiers for GWO with KL Divergence and IWO.

Feature Selection	Classifiers	Accuracy (%)	Error Rate (%)	F1 Score (%)	MCC	Jaccard Index (%)	g-mean (%)	Kappa
KL Divergence	SVM	67.72	32.28	71.40	0.37	55.52	68.80	0.35
	KNN	79.36	20.64	78.60	0.59	64.74	79.49	0.59
	Random Forest	75.78	24.22	74.24	0.52	59.03	76.09	0.52
	Decision Tree	77.41	22.59	76.09	0.55	61.41	77.69	0.55
	Softmax Discriminant	75.64	24.37	80.08	0.57	66.77	80.74	0.51
	Multilayer Perceptron	67.19	32.81	69.49	0.35	53.25	67.54	0.34
	Bayesian LDC	62.27	37.73	64.72	0.25	47.84	62.49	0.25
IWO	SVM	76.63	23.37	79.02	0.55	65.31	77.82	0.53
	KNN	70.83	29.17	68.71	0.42	52.34	71.16	0.42
	Random Forest	71.75	28.25	75.87	0.46	61.12	74.14	0.43
	Decision Tree	67.87	32.13	72.18	0.38	56.47	69.50	0.36
	Softmax Discriminant	69.06	30.94	73.17	0.40	57.69	70.74	0.38
	Multilayer Perceptron	71.22	28.78	75.25	0.45	60.32	73.33	0.42
	Bayesian LDC	59.76	40.24	61.52	0.20	44.42	59.84	0.20

**Table 14 diagnostics-13-03289-t014:** Performance Analysis of the Classifiers: PSO with KL Divergence and IWO for Adam Hyperparameter Tuning.

Feature Selection	Classifiers	Accuracy (%)	Error Rate (%)	F1 Score (%)	MCC	Jaccard Index (%)	g-mean (%)	Kappa
KL Divergence	SVM	76.56	23.44	77.72	0.53	63.56	76.80	0.53
	KNN	86.70	13.30	86.30	0.74	75.96	86.81	0.73
	Random Forest	80.75	19.25	82.56	0.63	70.30	81.86	0.62
	Decision Tree	76.24	23.76	76.26	0.52	61.63	76.24	0.53
	Softmax Discriminant	83.40	16.60	84.11	0.67	72.58	83.62	0.67
	Multilayer Perceptron	79.28	20.72	78.93	0.59	65.20	79.31	0.59
	Bayesian LDC	81.03	18.97	81.42	0.62	68.67	81.08	0.62
IWO	SVM	76.84	23.16	75.66	0.54	61.84	77.05	0.54
	KNN	84.33	15.67	83.82	0.69	72.14	84.44	0.69
	Random Forest	81.67	18.33	81.83	0.63	69.25	81.68	0.63
	Decision Tree	80.21	19.79	78.95	0.61	65.23	80.56	0.60
	Softmax Discriminant	82.90	17.10	82.84	0.66	70.71	82.90	0.66
	Multilayer Perceptron	85.64	14.36	86.28	0.72	75.88	85.94	0.71
	Bayesian LDC	76.14	23.86	76.19	0.52	61.54	76.14	0.52

**Table 15 diagnostics-13-03289-t015:** Performance Analysis of the Classifiers: PSO with KL Divergence and IWO for RAdam Hyperparameter Tuning.

Feature Selection	Classifiers	Accuracy (%)	Error Rate (%)	F1 Score (%)	MCC	Jaccard Index (%)	g-mean (%)	Kappa
KL Divergence	SVM	78.11	21.89	79.11	0.56	65.44	78.32	0.56
	KNN	87.45	12.55	87.02	0.75	77.03	87.58	0.75
	Random Forest	81.95	18.05	83.53	0.65	71.71	82.92	0.64
	Decision Tree	78.09	21.91	78.19	0.55	64.19	78.10	0.54
	Softmax Discriminant	85.15	14.85	85.61	0.70	74.84	85.27	0.70
	Multilayer Perceptron	82.09	17.91	81.75	0.64	69.13	82.12	0.64
	Bayesian LDC	82.92	17.08	83.25	0.66	71.31	82.96	0.66
IWO	SVM	79.45	20.55	78.31	0.59	64.35	79.72	0.59
	KNN	85.83	14.17	85.59	0.72	74.81	85.86	0.72
	Random Forest	84.17	15.83	84.23	0.68	72.76	84.17	0.68
	Decision Tree	82.21	17.79	81.08	0.65	68.18	82.58	0.64
	Softmax Discriminant	84.40	15.60	84.43	0.69	73.05	84.40	0.69
	Multilayer Perceptron	86.32	13.68	86.95	0.73	76.91	86.59	0.73
	Bayesian LDC	78.14	21.86	78.29	0.56	64.21	78.14	0.56

**Table 16 diagnostics-13-03289-t016:** Performance Analysis of the Classifiers: GWO with KL Divergence and IWO for Adam Hyperparameter Tuning.

Feature Selection	Classifiers	Accuracy (%)	Error Rate (%)	F1 Score (%)	MCC	Jaccard Index (%)	g-mean (%)	Kappa
KL Divergence	SVM	81.19	18.81	79.53	0.63	66.02	81.88	0.62
	KNN	90.87	9.14	90.06	0.83	81.92	91.71	0.82
	Random Forest	79.56	20.44	79.83	0.59	66.43	79.58	0.59
	Decision Tree	76.20	23.81	75.33	0.53	60.43	76.30	0.52
	Softmax Discriminant	78.62	21.38	76.93	0.58	62.51	79.13	0.57
	Multilayer Perceptron	77.74	22.26	77.07	0.56	62.69	77.82	0.55
	Bayesian LDC	77.62	22.38	77.14	0.55	62.78	77.66	0.55
IWO	SVM	89.56	10.44	89.27	0.79	80.61	89.66	0.79
	KNN	84.50	15.51	84.19	0.69	72.70	84.54	0.69
	Random Forest	89.76	10.24	89.80	0.80	81.49	89.76	0.80
	Decision Tree	90.07	9.93	90.29	0.80	81.30	90.13	0.80
	Softmax Discriminant	84.13	15.87	84.75	0.68	73.54	84.31	0.68
	Multilayer Perceptron	83.21	16.79	84.00	0.67	72.42	83.47	0.66
	Bayesian LDC	87.27	12.73	88.19	0.75	78.88	88.00	0.75

**Table 17 diagnostics-13-03289-t017:** Performance Analysis of the Classifiers: GWO with KL Divergence and IWO for RAdam Hyperparameter Tuning.

Feature Selection	Classifiers	Accuracy (%)	Error Rate (%)	F1 Score (%)	MCC	Jaccard Index (%)	g-mean (%)	Kappa
KL Divergence	SVM	82.24	17.76	80.89	0.65	67.91	82.77	0.64
	KNN	91.07	8.94	90.30	0.82	82.31	91.61	0.82
	Random Forest	81.56	18.44	81.62	0.63	68.95	81.56	0.63
	Decision Tree	77.35	22.66	76.80	0.55	62.34	77.39	0.55
	Softmax Discriminant	79.67	20.33	78.31	0.60	64.35	80.05	0.59
	Multilayer Perceptron	79.44	20.56	78.97	0.59	65.25	79.49	0.59
	Bayesian LDC	80.42	19.58	79.96	0.61	66.61	80.47	0.61
IWO	SVM	91.06	8.94	90.86	0.82	83.24	91.13	0.82
	KNN	85.50	14.51	85.36	0.71	74.46	85.50	0.71
	Random Forest	90.26	9.74	90.35	0.81	82.39	90.27	0.81
	Decision Tree	91.57	8.43	91.71	0.83	84.70	91.87	0.83
	Softmax Discriminant	85.13	14.87	85.71	0.70	75.00	85.32	0.70
	Multilayer Perceptron	85.21	14.79	85.77	0.71	75.09	85.39	0.70
	Bayesian LDC	88.62	11.38	89.38	0.78	80.80	89.25	0.77

**Table 18 diagnostics-13-03289-t018:** Performance Analysis of the classifiers for Maximum Accuracy.

S No	Feature Extraction	Feature Selection	Classifiers	Accuracy (%)
1	PSO	-	Bayesian LDC	83.42%
2	GWO	-	Decision Tree	85.01%
3	PSO	KL Divergence	Softmax Discriminant	83.47%
4	PSO	IWO	KNN	74.74%
5	GWO	KL Divergence	KNN	79.36%
6	GWO	IWO	SVM	76.63%
7	PSO	KL Divergence	KNN with Adam	86.70%
8	PSO	IWO	MLP with Adam	85.64%
9	PSO	KL Divergence	KNN with RAdam	87.45%
10	PSO	IWO	MLP with RAdam	86.32%
11	GWO	KL Divergence	KNN with Adam	90.87%
12	GWO	IWO	Decision Tree with Adam	90.07%
13	GWO	KL Divergence	KNN with RAdam	91.07%
14	GWO	IWO	Decision Tree with RAdam	91.57%

**Table 19 diagnostics-13-03289-t019:** Computational Complexity of the classifiers among Feature Extraction, Feature Selection and Hyperparameter Tuning approaches.

S No	Classifiers	Without Feature Extraction	With Feature Extraction		With Feature Selection		With Hyperparameter Tuning of IWO Feature Selection Method	
			PSO	GWO	KL Divergence	IWO	Adam	RAdam
1	SVM	$O\;({2n}^{2})$	$O\;({2n}^{5})$	$O\;({2n}^{5})$	$O\;({2n}^{6})$	$O\;({2n}^{6}\log n)$	$O\;({2n}^{2}\log n)$	$O\;({4n}^{7}\log5n)$
2	KNN	$O\;(n^{2})$	$O\;(n^{5})$	$O\;(n^{5})$	$O\;(n^{6})$	$O\;(n^{6}\log n)$	$O\;({2n}^{7}\log2n)$	$O\;({2n}^{7}\log5n)$
3	RF	O (nlog⁡n)	$O\;(n^{4}\log n)$	$O\;(n^{4}\log n)$	$O\;(n^{5}\log n)$	$O\;(n^{5}\log2n)$	$O\;({2n}^{6}\log3n)$	$O\;({2n}^{6}\log6n)$
4	DT	O (log⁡n)	$O\;(n^{3}\log n)$	$O\;(n^{3}\log n)$	$O\;(n^{4}\log n)$	$O\;(n^{4}\log2n)$	$O\;({2n}^{5}\log3n)$	$O\;({2n}^{5}\log6n)$
5	SDC	$O\;(n^{2})$	$O\;(n^{5})$	$O\;(n^{5})$	$O\;(n^{6})$	$O\;(n^{6}\log n)$	$O\;({2n}^{7}\log2n)$	$O\;({2n}^{7}\log5n)$
6	MLP	$O\;(n^{5})$	$O\;(n^{8})$	$O\;(n^{8})$	$O\;(n^{9})$	$O\;(n^{9}\log n)$	$O\;({2n}^{10}\log2n)$	$O\;({2n}^{10}\log5n)$
7	BLDC	$O\;(n^{2})$	$O\;(n^{5})$	$O\;(n^{5})$	$O\;(n^{6})$	$O\;(n^{6}\log n)$	$O\;({2n}^{7}\log2n)$	$O\;({2n}^{7}\log5n)$

**Table 20 diagnostics-13-03289-t020:** Comparison of classifier performance with different datasets.

S No	Authors	Dataset Used	Machine Learning Models/Classifiers	Accuracy (%)
1	Bukhari, S. et al. [57]	CRAG Dataset	ResNet-50	93.91%
2	Jinsa Kuruvilla. et al. [58]	LIDC Dataset (155 Patients—CT images)	Feed Forward Back Propagation Neural Networks	93.3%
3	Dabass, M. et al. [59]	CRAG Dataset	Atrous Convolved Hybrid Seg-Net Architecture	87.63%
4	Supriya, Suresh.et al. [60]	LIDC-IDRI Repository (CT scans)	CNN	93.9%
5	Wadood, Abdul. [61]	LIDC-IDRI Repository (CT scans)	CNN-ALCDC	97.2%
6	Rekka, Mastouri. et al. [62]	LUNA16 Database (3186 CT images)	BCNN [VGG16, VGG19]	91.99%
7	Tasnim, Ahmed. et al. [63]	LUNA16 Database	3D CNN	80%
8	Mesut Toğaçar. et al. [64]	Cancer Imaging Archieve (CT images)	AlexNet and kNN classifier	98.74%
9	Anum, Masood. et al. [65]	Biomedical Datasets–IoT	CNN DFCNet	77.6% 84.58%
10	Wahyudi, Setiawan. et al. [66]	LC25000 Database	CNN	87.16%
11	Manaswini, Pradhan. [67]	LC25000 Database	Without Feature Selection (EGOA)—KNN With Feature Selection (EGOA)–KNN	80.16% 81.59%
12	Phankokkruad, M [68]	LC25000 Database	Ensemble ResNet50V2	91% 90%
13	Karthikeyan Shanmugam, Harikumar Rajaguru	LC25000 Database	Feature Extraction-GWO Feature Selection—IWO Decision tree with RAdam Hyper parameter Updation method	91.57%

## Data Availability

Not Applicable.

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
