# Peer review of "Exploration and Enhancement of Classifiers in the Detection of Lung Cancer from Histopathological Images"

_diagnostics, 2023, doi:10.3390/diagnostics13203289_

Round 1

Reviewer 1 Report

This paper is an interesting approach to the analysis of medical images. Nevertheless, in methods based on learning or classification, an important element is the selection of appropriate data. The effectiveness of forecasting and the thesis depends largely on the selection of criteria and the amount of data.

Minor remarks:

1) I propose to describe in more detail on what basis and what results from the scope of selection of input data for the research problem.

2) The authors could refer to other methods and justify the choice of the presented solutions.

3) What is unique in the presented research compared to other work on the given problem?

Author Response

Dear sir,

We  are submitting the revised  manuscript  titled,  “Exploration  and  Enhancement  of  Classifiers in the Detection  of  Lung  Cancer  from  Histopathological  Images”  after incorporating the answers to the Reviewers comments in the Highlighted form.

This is for your kind information and perusal.

With Regards,

Harikumar Rajaguru

Reviewer 2 Report

We thank the authors for addressing our previous concerns.

N/A

Author Response

Dear sir,

Thank you for your review and valuable suggestions.

With Regards,

Harikumar Rajaguru
